# DESIGN IN THE DARK: LEARNING DEEP GENERATIVE MODELS FOR DE NOVO PROTEIN DESIGN

## ABSTRACT

The design of novel protein sequences is providing paths towards the development of novel therapeutics and materials. At the forefront is the challenging field of *de novo* protein design, which looks to design protein sequences unlike those found in nature using general design methodologies. In this work, we develop a tool for *de novo* design, based on a deep generative sequence model, that rapidly samples novel protein sequences with diverse and ordered structures. To build this tool we develop a framework, called DARK, that trains the underlying generative model on an iteratively expanding set of synthetic sequences. The resulting model generalizes where models trained on natural sequences struggle and greatly improves on the efficiency of comparable sampling-based approaches. We further show how it can generate high quality candidates for *de novo* design problems and aid in the development of further novel design methods, in all, providing another step, amongst others, towards truly automated and intelligent protein design.

## 1 INTRODUCTION

Generative modelling is beginning to be used for the task of designing protein molecules, a problem which offers potential solutions to a vast number of medical (Chevalier et al., 2017; Silva et al., 2019) and industrial challenges (King et al., 2012; Wang et al., 2021). Computational protein design methods look to efficiently generate large numbers of candidate sequences, prior to laboratory validation, that are confidently predicted to have stable and ordered structures together with pre-specified structural and functional attributes. Recent generative modelling studies have looked to improve on the high time and computational cost of contemporary design methods, which rely on Monte Carlo sampling and energy function-based physics simulations (Huang et al., 2016).

Here, our aim is to develop a tool, based on a deep generative model, that can rapidly generate sequences with stable and ordered structures. Furthermore, we build this tool for the important task of *de novo* protein design. Computational *de novo* protein design is arguably the most promising and general approach to protein design, but also the most difficult (Huang et al., 2016; Korendovych & DeGrado, 2020). As an approach, it looks to design proteins unlike those seen in nature, with desired structural or functional attributes. Crucially, this is done without using any information from a pre-existing protein as a starting point, scaffold, or guide.[1] Instead, the designer must rely on general design methods with little, if any, provided information to design suitable sequences with suitable predicted structures (Woolfson, 2021; Marcos et al., 2018; Vorobieva et al., 2021). This presents a remarkably challenging design setting, evident in that *de novo* designs have yet to be successfully generated for a number of common but biotechnologically important folds, like immunoglobins.

A tool that quickly generates sequences, unlike those seen in nature, with a diverse range of ordered structures is of great value as it can be used to both rapidly generate candidates directly for design tasks as well as provide a basis for developing further novel design methods. In this work, we develop deep generative sequence models that we show as satisfying this specification. Unlike previous generative protein design approaches that typically rely on natural sequences and structure-based conditioning information, we build an approach for *de novo* design by developing a framework for training unconditional autoregressive language models on synthetic sequences. We call

---

[1] The opposite of such an approach is protein engineering, an important but distinctly different field we do not cover here. However, for clarity, an example of protein engineering in a machine learning context would be fine-tuning a pre-trained language model on existing proteins with the desired attributes.

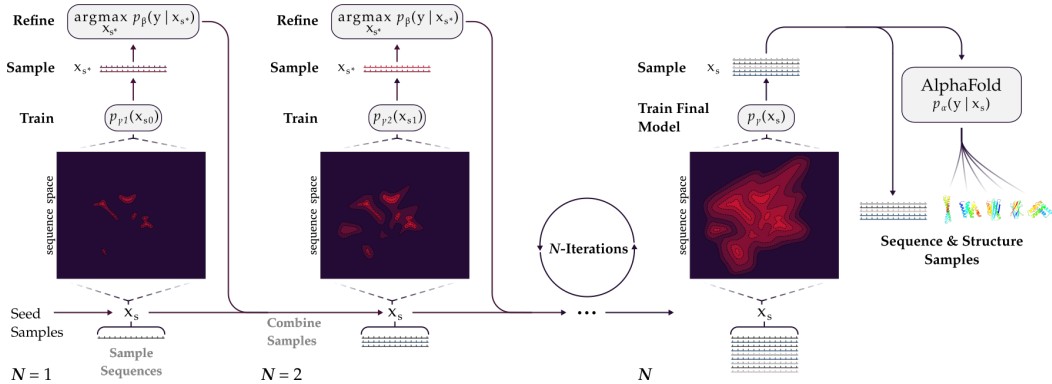

Figure 1: **Growing a synthetic dataset and training unconditional language models with DARK.** An illustrative overview of the procedure used by DARK to iteratively grow a training set of synthetic sequences that are used to train increasingly performant generative models. The final structure generation with AlphaFold for sampled sequences is also included.

this framework Design in Areas of Restricted Knowledge (DARK) [2]. DARK iteratively generates large datasets of synthetic sequences that are optimized in an unsupervised way to have unspecified ordered structures that are highly likely under a supervised protein structure predictor (Figure 1). We build this tool as an unconditional model to avoid limiting its potential applications and also to show that it is possible to learn general relationships between sequence and structure without the aid of conditioning information.

We evaluate our approach by first demonstrating that DARK models satisfy our criteria for a *de novo* design tool. In particular, we show that the final model, DARK$_3$, generates novel sequences with ordered structures as judged by state-of-the-art structure predictor AlphaFold (V2; Jumper et al. (2021)). We also test the *structural generalization* of our approach using a stringent unseen structure-based test set and find DARK models performs well. We demonstrate the applications of DARK$_3$ as our tool with an example of designing a sequence with fold commonly used as a *de novo* scaffold for grafting functional sites. Finally, we show that DARK models enable the development of new methods by developing AlphaFold refinement, a novel and efficient approach for producing high confidence design candidates with AlphaFold. We also use it to de novo design a sequence with a high confidence immunoglobin fold.

In summary, we make the following contributions: **(1)** We show that unconditional generative models of protein sequences can learn distributions that capture general structure information by learning from synthetic sequences. **(2)** We propose a novel framework, DARK, for efficiently training deep generative models on synthetic protein sequences. **(3)** We provide the final DARK$_3$ model as a flexible and fast tool for *de novo* design that generates novel sequences with diverse ordered structures. **(4)** We demonstrate AlphaFold refinement, a novel way to efficiently *de novo* design high confidence sequence candidates using AlphaFold and DARK.

## 2 RELATED WORK

Here, we discuss related work applying generative models to protein design as well as those that have focused on *de novo* design, and broader contemporary protein design methods.

**Generative sequence modelling in protein design**     There have been a variety of machine learning approaches (Killoran et al., 2017; Wang et al., 2018; Norn et al., 2021) including generative modelling (Anand & Huang, 2018; Sabban & Markovsky, 2020; Linder et al., 2020) in protein design and protein engineering (Yang et al., 2018; Sinai & Kelsic, 2020). Unconditional and conditional generative models have been explored in a variety of problems adjacent to protein design, such as variant prediction (Riesselman et al., 2018; Shin et al., 2021). Other examples are recent work using

---

[2]This is inspired by the huge 'dark' areas of the sequence space that are unexplored by nature (Taylor et al., 2009; Perdigão et al., 2015).

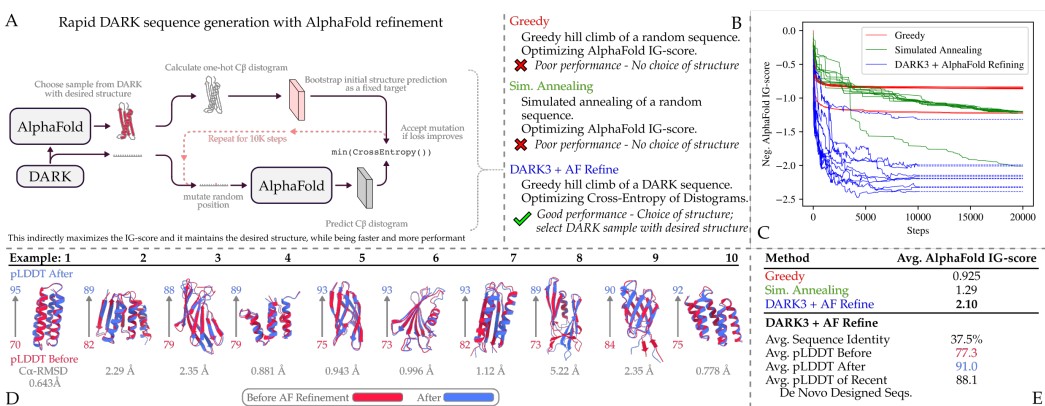

Figure 2: **Generating *de novo* designed sequences with AlphaFold refinement and DARK₃.** (A) Overview of our AlphaFold refinement approach. (B) Comparisons of the method and two benchmarks in C. (C) Plot showing the design trajectories for 10 DARK samples with AlphaFold refinement (10K steps) compared to 10 runs of the Simulated Annealing benchmark and 10 runs of the Greedy benchmark (both 20K steps).(D) Structures and metrics of the 10 AlphaFold refinement examples in C before and after refinement. (E) Averaged metrics of the benchmarked methods, with more detailed metrics for AlphaFold refinement.

language models to learn representations for downstream supervised tasks (Alley et al., 2019; Bepler & Berger, 2019; Elnaggar et al., 2020; Rao et al., 2021).

Most relevant to the work presented here is the set of models that have been developed to explore design using generative models conditioned on structural information. Although a range of architectures have been used, these models vary primarily in the representations of structural information used for conditioning. Amongst others, models have conditioned on low information descriptions of protein folds (weak conditioning) (Greener et al., 2018; Karimi et al., 2020), secondary structure labels (Singer et al., 2021), and course descriptions of protein backbones (Ingraham et al., 2019; Strokach et al., 2020; Cao et al., 2021).Some methods train proteins of one structural or functional type, effectively conditioning on that specific type (Yu & Buehler, 2020; Shin et al., 2021).

Conditioning out structure provides a way to address an inherent challenge to working with natural protein sequences. The vast majority of known natural proteins have either a weak signal for structure or it is mixed in with other information, such as functional constraints and evolutionary drift. Unconditional generative models, trained on natural sequences, fail to generalize to sequence with structures beyond those represented by their training sets because of this lack of structure signal (Ingraham et al., 2019). This is also well understood when viewed from the lens of protein structure prediction. Even highly accurate structure predictions models like AlphaFold (Jumper et al., 2021) rely on the information contained in aligned sets of sequences similar to the query sequence to achieve accurate predictions for natural sequences. With only individual natural sequences, performance tends to be poor (Moffat & Jones, 2021; Xu et al., 2021).

**Generative Models and *de novo* protein design**    A generative sequence model for *de novo* design needs to be highly general, learning general sequence and structure relationships. All considered, we find it unsurprising that there has been no clear and unambiguous demonstration of a deep generative model, without any aid from conditioning, that generates sequences unlike natural sequences and, crucially, with confidently predicted ordered structures. Here, we demonstrate such a model. We achieve this by learning from syntheticsequences with a strong structure signal.

Previously *de novo* designed sequences are known to have extremely strong structure signals, but just over 100 *de novo* designed sequences (Woolfson, 2021) have lab-determined structures in the Protein Data Bank (PDB) (Burley et al., 2021). Thus, we find an alternative approach which is to generate synthetic sequences by leveraging a sequence optimization objective from the recent trDesign method (Anishchenko et al., 2020). Like contemporary approaches (Leaver-Fay et al., 2011), it is primarily a Monte Carlo sampling method, however in contrast, it slowly samples from a trained supervised structure predictor, using simulated annealing to maximize the structure likelihood of random sequences. The final sequences were shown to be extremely similar to pre-existing *de novo* designed sequences in their attributes and to have a strong structure signal.

**Contemporary *de novo* protein design** Although recent nonparametric methods have been developed (Zhou et al., 2020), by far the most historically successful *de novo* design approach is the Rosetta framework (Leaver-Fay et al., 2011), which is the recognized gold-standard (Huang et al., 2016). The Rosetta approach constitutes a complex computational pipeline that uses a variety of different protocols to iteratively sample viable structures and sequences using physics-based force-field simulations and Monte Carlo search procedures (Vorobieva et al., 2021; Marcos et al., 2018) making it lengthy and computationally expensive. New techniques that can rapidly produce quality samples, such as the work presented here, offer the potential to greatly reduce these costs. For further discussion regarding *de novo* design, we point the reader to the following reviews and retrospectives by Huang et al. (2016), Korendovych & DeGrado (2020), and (Woolfson, 2021).

## 3 LEARNING DARK MODELS

### 3.1 BACKGROUND

Let $x \in V^L$ be a protein sequence of length $L$ over a vocabulary $V$ of discrete amino acids ($|V| = 20$). Each amino acid is textually represented by a one-letter code. Each sequence $x$ maps to a corresponding molecular structure $y$ in three-dimensional space. Protein structure is described by the 3D coordinates of each atom, however more coarse-grained representations are commonly used. Here, we use the discrete representation of structure known as distograms (Senior et al., 2020). When using AlphaFold we use its predicted distograms and atomic coordinates, depending on the task (Jumper et al., 2021). Distograms represent a structure as a $[L, L, D]$ tensor, comprising $D$-dimensional one-hot encodings of a discretized distance measure between residue pairs. We do not consider the amino acid Cysteine for experimental reasons when generating synthetic sequences[3], however, for consistency we allow all trained models to predict it. For simplicity, we assume a fixed length of $L = 100$, however our approach easily generalizes to variable sequence lengths as it is based on autoregressively factorized probability models.

We are interested in the set of sequences that are unlike those in nature $x_d \in V^L$ with structures that are folded in some ordered and regular state $y_d$. We would like to sample from their joint distribution, which is convenient to break down, by chain rule, to $p(x_d, y_d) = p(y_d|x_d)p(x_d)$, a common way to consider the protein design problem. The first term $p(y_d|x_d)$ is often thought of as protein structure prediction and so it is convenient to model it with some accurate proxy regression model $p_\beta(y_d|x_d)$, being a trained supervised structure predictor, with parameters $\beta \in B$. We refer to this proxy regression model as an *oracle*. In this work, we focus on modeling the prior, $p(x_d)$ as the tool we desire, which, when combined with AlphaFold, provides a rapid means to produce high quality sequence and structure samples for *de novo* design problems.

### 3.2 AN UNSUPERVISED DESIGN LOSS FUNCTION

Recently, Anishchenko et al. (2020) demonstrated a method of sampling sequences for *de novo* design which, using our own notation, also looks to maximize $p(y_d|x_d)p(x_d)$, using the trRosetta structure predictor as an oracle (Yang et al., 2020). This samples sequences using simulated annealing of random sequences to optimize a combined proxy objective for both terms, ultimately generating sequences with the structure signal we are interested in. Here we describe the proxy objective in our own notation and then how we use it. First, $p(y_d|x_d)$ is approximated using the Kullback-Leibler divergence ($D_{\mathrm{KL}}$) between an oracle and a trained *background* network $p_\epsilon(y_d)$ which approximates a prior over ordered structures.

$$\arg\max_{x_d} \log p_\beta(y_d|x_d) \approx \arg\max_{x_d} D_{\mathrm{KL}}\Big(p_\beta(y_d|x_d)\|p_\epsilon(y_d)\Big) \tag{1}$$

The background network is trained by taking a model with the same architecture and dataset as $p_\beta(y_d|x_d)$ and training it with the sequence inputs replaced by Gaussian noise[4] (Appendix A.1). The final objective also includes a term for approximating a weak sequence prior by taking the

---

[3]This is common in design studies, in part because it significantly complicates otherwise straightforward recombinant protein expression in bacteria, the most relevant factor for us (See Appendix A.7).

[4]In practice, for a given length, the background model distograms can be calculated once, taking an average over an arbitrary number of predictions (e.g. 100), and then saved to be used for all IG-score calculations.

negative $D_{\mathrm{KL}}$ between the sequence's normalized unigram frequencies $p(x_d^{uni})$ and the unigram frequencies of proteins with high resolution crystal structures in the PDB, $p(x_{PDB})$. For simplicity, we refer to this objective as the Information Gain score (IG-score):

$$\text{IG-score} = \underset{x_d}{\arg\max} \left( D_{\mathrm{KL}}\Big(p_\beta(y_d|x_d)\|p_\epsilon(y_d)\Big) - D_{\mathrm{KL}}\Big(p(x_d^{uni})\|p(x_{PDB})\Big) \right) \qquad (2)$$

Here, we implement our own version of trDesign, termed $\text{DARK}_0$, and correspondingly our own version of the IG-score, using a version of the recent DMPfold2 model, chosen for its fast inference, for the oracle and background network (Kandathil et al., 2020), which is modified so that it produces distograms instead of atomic coordinates (Appendix A.1). $\text{DARK}_0$ is slow, it takes an average of 11.4 minutes to generates a sample. Given the end point of protein design is in the lab, a performant design model needs to provide as many high confidence candidates as possible to justify the cost of validation as well as combat very high attrition rates. A method like trDesign is too slow, and has less potential future adaptions, to be used as the tool we desire. Instead we show how we can use IG-score to learn generative models that sample similar sequences orders of magnitude faster with many potential applications. To kickstart the approach described in the next section, we use $\text{DARK}_0$ (40,000 simulated annealing steps) to sample a small number of sequences, 15K, that we call the *seed samples*. It is also used to sample two sets of 950 sequences to be used as a validation and test set for models in the next section (See Appendix B.1 for further details). After this $\text{DARK}_0$ is discarded as an approach, and the only aspect of trDesign that we build on is our own version of the IG-score; the rest of what is described here is our own construction.

### 3.3 BUILDING THE DARK FRAMEWORK FOR DESIGN

---

**Algorithm 1** Design in Areas of Restricted Knowledge (DARK)

---

**Input:** Oracle model $p_\beta(y_d|x_d)$
**Input:** Iterations $N$
**Input:** Sample sizes at each iteration $m_{0:N} = \{m_0, ..., m_N\}$
1:   $x_{s^0} \leftarrow$ Generate $m_0$ seed samples from $p_\beta(y_d|x_d)$
2:  **for** $n = 1, 2, ..., N$ **do**
3:      $p_{\gamma^n}(x_s) \leftarrow p_{\gamma^n}(x_s|x_{s^{n-1}})$, Learn a new sequence model
4:      $x_{s*} \leftarrow$ Ancestral sampling of $p_{\gamma^n}(x_s)$                       ▷ $m_n$ new samples
5:      $x_{s*} \leftarrow$ Rapid optimization of $x_{s*}$ IG-score using the oracle
6:      $x_{s^n} \leftarrow [x_{s*}, x_{s^{n-1}}]$ Combine optimized samples with existing samples
7:  **end for**

---

DARK (Algorithm 1) tackles the primary limitation on effectively learning models, which is having suitable sequences to train and test against. The first step is to take some initial set of samples $x_{s^0}$ to train an initial sampling model $p_{\gamma^1}(x_{s^0})$ with parameters $\gamma^1 \in \Gamma$. Here we use the aforementioned seed samples, but this could be random sequences or sequences from $\text{DARK}_0$-*Grad*, an approach we developed that backpropagates the negative IG-score to generate samples (Appendix A.2), in a fashion similar to (Killoran et al., 2017). We use the seed samples out of convenience as they have the highest score, and we reasoned that they would require less iterations by comparison. The sampling models are all generative sequence models that can be sampled. After training $p_{\gamma^1}(x_{s^0})$, it is used to generate a large set of samples $x_{s*}$. Instead of then training a new sampling model on $x_{s*}$, each sequence in $x_{s*}$ is quickly optimized to improve the IG-score.

This is done by a quick greedy hill-climb on the sequence for a set number of steps (Line 5 in Algorithm 1), which we refer to as *refinement*. In each step, a random position is mutated. If it improves the IG-score then it is kept, otherwise it is discarded. We use 3000 steps, a fraction of the steps done in $\text{DARK}_0$, as this was found to improve the average IG-score of $x_{s*}$ to approximately equal to the IG-score of $x_{s^0}$. For simplicity, this is fixed for all steps. The result of optimizing $x_{s*}$ is then combined with $x_{s^0}$ to make $x_{s^1}$. In the second iteration, $x_{s^1}$ is used to train a new sampling model $p_{\gamma^2}(x_{s^1})$. After some number of iterations $N$, the final sampling model is viewed as a strong prior over $x_d$, also making it a powerful generative model that can be used for *de novo* design tasks. Here we perform 3 iterations with training set sizes of 15K, 100K, and 500K, where the difference is sampled and refined between iterations. These sample sizes are arbitrary and they were entirely dictated by computational resources available at the time. Three iterations were performed

Table 1: **Perplexities (PPLs), IG-scores, & Sampling Speed for protein language models.** Lower PPL is better and higher IG-score is better. IG-scores are calculated from 1000 sequence samples except the 15K $DARK_0$-Seed Samples. IG-scores for refined (Ref) samples are included for DARK models. Sampling Speed is for a single example (See Appendix A.6 for further details).

| Language Model | PPL Test ↓ | IG-score ↑ | IG-score (Ref) ↑ | Sample Speed (s) ↓ |
|---|---|---|---|---|
| Unigram | 17.69 | 0.27 | - | - |
| Pfam HMM profiles* | 11.64* | - | - | - |
| UF50 | 12.11 | 0.21 | - | 0.0524 |
| LSTM | 9.94 | 0.92 | - | 0.0210 |
| $DARK_1$ | 6.33 | 1.38 | 1.77 | 0.0234 |
| $DARK_1$-*Adversarial* | 6.26 | 1.33 | - | 0.0234 |
| $DARK_2$ | 5.73 | 1.53 | 1.81 | 0.0524 |
| $DARK_2$-*STRICT* | 6.50 | 1.48 | - | 0.0524 |
| $DARK_3$ | **5.20** | **1.63** | **1.84** | 0.0524 |
| $DARK_0$-*Seed Samples* | - | 1.76 | 1.76 | 685 |
| $DARK_0$-*Grad* | - | 0.97 | - | 15.1 |

*Values reproduced from Ingraham et al. (2019); a conditional model conditioned on a specific sequence position of a specific family.

as we believed it sufficient to demonstrate DARK's effectiveness. We note that the iterative aspect of DARK shares similarities to methods like Estimation of Distribution Algorithms (EDAs), a common approach in model based optimization (MBO) (Bengoetxea et al., 2001; Larrañaga & Lozano, 2001; Brookes et al., 2020). Viewed through this lens, it suggests a number of ways that DARK can be adapted and potentially improved on, which we leave for future work (Kumar & Levine, 2020; Trabucco et al., 2021).

**Deep generative architectures** In DARK, we use deep autoregressive language models for learning protein sequence distributions. Specifically, we use a standard Transformer decoder architecture (Vaswani et al., 2017; Radford et al., 2019) which has been used extensively to train state-of-the-art language models capable of generating high-quality synthetic sequence samples. In the first iteration of DARK we use a small decoder model, termed $DARK_1$, with 4 layers, 4 heads, and a feed-forward size of $H = 128$. To affirm the choice of self-attention based architecture, we compare it to a 1-layer LSTM with a hidden dimension $H = 128$. We also include results for a variant of $DARK_1$ referred to as $DARK_1$-*Adversarial*, that uses DMPfold2 as an adversarial regularizer during training (Appendix A.3). After the first iteration, the second and third iteration models, $DARK_2$ and $DARK_3$ respectively, use a decoder with 12 layers, 12 heads, and $H = 768$. The parameters are increased to coincide with the order of magnitude change in the size of the training set from $DARK_1$ to $DARK_2$. To provide a even comparison to models trained on natural sequences, we train a model, termed UF50, on 50M natural sequences from the UniRef50 sequence database (The UniProt Consortium, 2021) using the same architecture as $DARK_3$ (Appendix A.5). For all trained models we perform early stopping and a small amount of hyper-parameter optimization on the validation set.

**Quantitatively measuring sample quality** We define high-quality samples as those that are diverse, have sequences unlike any natural sequence, and are predicted to have a stable and ordered structure. In *de novo* design, the most important test of a candidate sequence is that it is confidently predicted to have a stable and ordered atomistic structure (Huang et al., 2016). We provide a direct measure of both confidence and order jointly using AlphaFold's pLDDT score, being its confidence metric, which ranges from 0 to 100 (Jumper et al., 2021). This is a measure of confidence, and an indirect measure of order. It was found that low scores (pLDDT< 50) strongly suggest disorder and vice versa (Tunyasuvunakool et al., 2021). Thus, we use the proportion of samples with a pLDDT> 70 as a measure of high quality for the predicted structures, which we refer to as *Good+ pLDDT*, indicating "good or better" quality according to criteria in Tunyasuvunakool et al. (2021). When discussing results we use 'ordered' to mean 'confidently predicted and ordered'. Using AlphaFold also provides a check to ensure that the model has not learned to produce adversarial examples against the original DMPfold2 oracle. We also measure the diversity of sequences and predicted structures using the estimated number of clusters in samples of sequences, and samples of predicted structures (See Appendix B.7 & B.8).

Table 2: **DARK$_3$ produces samples with confidently predicted structures by AlphaFold.** Good+ pLDDT scores from AlphaFold (higher is better) are shown for unconditional language models. See Table 8 for additional benchmarks.

| Language Model | Good+ pLDDT (%) $\uparrow$ | Training Data |
|---|---|---|
| Random | 1.8 | - |
| UF50 | 15.9 | Natural |
| DARK$_3$ | **32.3** | Synthetic |
| DARK$_0$-*Seed Samples* | 40.8 | Synthetic |

**Measuring the structural generality of DARK in a strict setting** We take two key approaches to providing a stringent measure of how well DARK generalizes with regards to structure. Both are enabled by using AlphaFold to generate all-atom structure predictions for all 100,000 sequences contained in the DARK$_2$ training set, which covers the 15,000 seed samples and the 85,000 samples from the first iteration of DARK. We then assign each sample's structure to a topology. This process is the repeated for the 950 sequence validation set; we ignore the 950 sequence test set for what is described here. The first approach we take is to carefully construct a *strict* training and test set split based on structure, from the 100K samples (See Appendix B.2 for exhaustive details). This has been recently suggested for machine learning-based protein design studies (Ingraham et al., 2019) and is considered a gold-standard approach in protein structure prediction (Chothia & Lesk, 1986; Söding & Remmert, 2011). Our strict test set contains no sequences with overlapping topologies in either the training set or validation set. We also remove any sequences from the validation and training set that are detected to be similar to those in the test set with the MMseqs2 search tool (Steinegger & Söding, 2017). We train and evaluate a model on these sets, termed DARK$_2$-*STRICT*. In the limit of working with synthetic, we believe this constitutes a stringent test of generality. The second approach we take is to compare the number of unique topologies represented by the 15,000 seed samples and those represented by the 85,000 sequences from the first iteration of DARK to show that DARK iterations generate sequences with unseen structures.

### 3.4 USING DARK$_3$ FOR DE NOVO DESIGN

**Rapid de novo design with DARK** We provide an example of how DARK can be used in a specific design task by setting the challenge of *de novo* designing a 4-helix bundle, which is commonly seen in design studies and in nature (Orengo et al., 1997). 4-helix bundles are also of interest as they are being used as *de novo* scaffolds for grafting in functional sites (Woolfson, 2021). To design a sequence, we use DARK and AlphaFold to generate a database of samples and predicted structures. We take the highest pLDDT 4-helix bundle from the dataset and then validate its structure prediction by predicting its structure using the Rosetta Abinitio Relax protocol without any homology information (Leaver-Fay et al., 2011) (this took 433 CPU hours). We generate 10,000 structure predictions (decoys), and rank them by lowest Rosetta all-atom score. For this example, we use the set of 1000 unrefined sequences from DARK$_3$ as if it were a structure database, but this can be expanded to an arbitrary size in practice. 1000 sequences with predicted structures is equivalent to generating 28 sequences with an approach like trDesign by time. Due to its speed, DARK is able to provide a large number of potential candidates sequences for different design tasks like this one in an off-the-shelf way.

**An novel but simple approach to de novo designing with AlphaFold** To demonstrate that DARK models can be used in method development, we use DARK$_3$ to develop a simple design method we call AlphaFold refinement (Figure 2A & 2B). This provides a solution to the problem of having a candidate sequence with a structure of interest to a designer but it has low confidence. AlphaFold refinement functions by bootstrapping the initial predicted structure as a target for performing a hill-climb based optimization of the DARK$_3$ sampled sequence. More specifically, we minimize the Cross-Entropy between the AlphaFold predicted distogram and the one-hot distogram of the bootstrapped target structure. To provide a comparison to AlphaFold refinement, we construct a trDesign-based approach that optimizes an AlphaFold IG-score (AIG-score) using simulated annealing. AIG-score also provides a measure of confidence. As an example of its practical use, we use AlphaFold refinement to *de novo* design a sequence with a high confidence immunoglobin fold (Figure 3) (See Appendix B.9 for exhaustive details).

## 4 RESULTS

Here, we assess DARK$_3$ as a tool and provide demonstrations of its use.

### 4.1 EVALUATING DARK MODELS AS A TOOL FOR DE NOVO PROTEIN DESIGN

**Comparing DARK to a simulated annealing approach**    We use DARK$_3$ without refinement as our final design tool and we find that it produces similar samples by IG-score to DARK$_0$ (seed samples), our version of trDesign, but does so four orders of magnitude faster, 52 ms compared to 11 mins per example (Table 1). We also find that the iterative approach to DARK effective for improving performance between iterations and significantly reduces the required resources. Rounding up to the nearest day, if we were to re-perform DARK from nothing to having a trained DARK$_3$ it would take 12 days when parallelized across ten V100 GPUs. Of that time, model training constitutes just over 3 days and only requires 1 GPU. To generate 500,000 examples with DARK$_0$ would take ~80 days across ten V100 GPUs . Compared to methods like trDesign, the sample speed of DARK models allows for the generation of large quantities of candidate sequences for different design tasks. With an approach like DARK's, that relies on generative neural networks, there are also clear ways to introduce controlled generation (Dathathri et al., 2019; Keskar et al., 2019).

**DARK generates samples with ordered structures**    We find that DARK$_3$ reliably generates samples with ordered predicted structures (Table 2). Without any refining or conditioning, 32.3% of DARK$_3$ samples have Good+ pLDDT scores. Given the IG-score of unrefined DARK$_3$ samples is slightly lower than that of the DARK$_0$ seed samples (Table 1), it is unsurprising that similar difference is present with the Good+ pLDDT (Table 2). Compared to an approach like trDesign, for a small decrease in Good+ pLDDT, DARK$_3$ is able to generate sequences orders of magnitude faster, also giving it much greater flexibility for integrating and developing future approaches.

**DARK samples are diverse**    We find that samples from DARK models have diverse sequences and predicted structures (Table 3 & 4). Across all iterations, clustering results suggest that only a small number of samples have similar sequences and structures. That said, clustering is not a very robust measure as it doesn't indicate an exact degree of diversity. We also generate 5000 sequences from DARK$_3$ and predict their structures to gain a further qualitative insight into the diversity of the structures represented in the samples. There appears to be a preference for all-$\alpha$ and all-$\beta$ structures, but $\alpha$ & $\beta$ proteins are still common. We include examples in Figure 5. Looking forward, future investigation is required to properly quantify and examine the structure distribution of sequences from the final model. That said, we do find sequences of design interest in the samples that we have generated. For example, sequences with topologies that have yet to be successfully de novo designed, like immunoglobin folds.

**DARK samples are unlike natural sequences**    We find that samples from the DARK$_3$ model are distinct from sequences in nature. A sequence search, using the MMseqs2 search tool (Steinegger & Söding, 2017) with a high sensitivity parameter (`-s 7`), against UniRef100 (The UniProt Consortium, 2021) of the 1000 evaluated sequence samples returned no hits for all sequences except one, which had 5 weak hits. Repeating the sequence search with a different tool, HHblits (Remmert et al., 2012), no natural sequences were found that met the HHblits default threshold value, being an E-value below $1 \times 10^{-3}$ (See Appendix B.7 for further details). This strongly suggests that DARK$_3$ generates sequences that are unlike those found in nature. We find the opposite results for the natural sequence model. Repeating the MMseqs2 search with the 1000 samples from the UF50 model results in 41 samples having a combined total of 1.76 million unique hits. Given the UF50 model is trained on natural sequences this is unsurprising. All considered, training on synthetic sequences appears to provide a simple means to generate sequences unlike those in nature.

**Learning from natural sequences leads to poor performance**    The UF50 model provides a direct comparison to having trained on natural sequences instead of synthetic sequences, and is similar to protein engineering focused approaches like ProGEN (Madani et al., 2020) and UniRep (Alley et al., 2019). As expected, it performs poorly by both perplexity and IG-score (Table 1) and underperforms compared to DARK$_3$ by Good+ pLDDT (Table 2). Especially given the test set comprises synthetic sequences, from a naive perspective we would expect UF50 to assign high likelihoods to the test set.

Not only is its training set two orders of magnitude larger than DARK$_3$'s, the test set sequences have a clear structure signal, It is also hyperparameter optimized on the synthetic validation set. For the Good+ pLDDT, it is also advantaged as many of the sequences in the AlphaFold training set are in its own training set. Ultimately, this result is entirely expected (See Section 2).

**DARK models generalize to sequences with unseen structures**   We find that DARK generalizes to sequences with unseen structures. DARK$_2$-*STRICT*, tested on on the strict test set (Section 3.3 & Appendix B.2), performs well and achieves a perplexity of 6.50, similar to the other DARK models and similarly outperforming the baselines (Table 1). In assigning high likelihoods to the sequences in the strict test set, DARK$_2$-*STRICT* provides direct evidence that DARK models are learning general relationships between sequences and structure. This is further supported by how unseen topologies are sampled between DARK iterations (Section 3.3). Within the 100K predicted structures of the DARK$_2$, we find that the initial 15K seed samples contains 421 unique topologies and the 85K set from the first iteration of DARK contains 1011 unique topologies, with an intersection of 373 between the two. As such, there are 638 topologies sampled during the first DARK iteration that were not present in the 15K training set. This accounts for 1476 sequences, of which 225 sequences (15.2%) have a pLDDT$\geq 70$. This provides direct evidence that the DARK iterations are expanding the training sets to include sequences with unseen and ordered structures, suggesting a general approach.

## 4.2 USING DARK$_3$ FOR DE NOVO DESIGN

**An example of *de novo* design with DARK**   For the task of designing a 4-helix bundle (Section 3.4), we find that DARK$_3$ successfully produces a candidate, effectively off-the-shelf. The predicted structure of the chosen sequence has a high pLDDT of 91.4, similar to the pLDDT of existing *de novo* proteins (Appendix B.4). We also find that its AlphaFold structure is in agreement with the best ranked predicted structure by Rosetta, having a low root-mean-square deviation of 3.11Å (by $\alpha$-carbons) compared to AlphaFold's prediction (Figure 6 & 8). In all, this provides a demonstration of how models like DARK$_3$, paired with AlphaFold, can rapidly provide *de novo* candidates for tasks like generating scaffolds for grafting in functional sites.

**Generating de novo designed sequences with AlphaFold refinement and DARK**   We find AlphaFold refinement can reliably generate high confidence (pLDDT 91.0) sequence candidates, while maintaining their structures,. This is slightly better than the confidence (pLDDT 88.1) with which AlphaFold predicts the structures of recently released *de novo* designed sequences in the PDB (Figure 2C & 2E). We also find that AlphaFold refinement is efficient, typically converging by 7500 steps which is $\sim$10 hours on a consumer-grade graphics card (RTX 2080 Ti). This is significantly more efficient than approaches like de novo design with Rosetta which can take thousands of CPU hours. In contrast, we found the simulated annealing approach produced what we consider an adversarial attack on AlphaFold (Figure 2E & 4). For the task of *de novo* designing a sequence with an immunoglobin fold, we find that DARK with AlphaFold refinement is also successful, producing a final predicted structure with a high pLDDT of 93.1 (Figure 3), demonstrating how DARK can be used to generate candidates for important but yet to be *de novo* designed folds.

## 5 CONCLUSION

In this work, we show that it is possible to build a tool, in the form of DARK$_3$, that can rapidly generate sequences with diverse and ordered structures. Looking forward, building a variable length training set and smoothing the topology distribution between iterations are clear first targets as well as exploring related approaches in MBO. Controlled generation is an especially useful route to explore and there is a clear way to approach it: AlphaFold structure predictions themselves can be used as pseudo-labels for training future conditional versions of DARK entirely based on synthetic sequences. The AlphaFold refinement approach can also be easily extended to the sub-problem of fixed-backbone protein design. Looking forward, we hope that DARK, and many others, are continuing to provide steps towards intelligent and truly automatic protein design.

ACKNOWLEDGMENTS

*Left blank for anonymity during double-blind peer review.*

REPRODUCIBILITY STATEMENT

In order to guarantee reproduciblity of this work, we provide a detailed appendix with specific implementation and methodological details which is sufficient to reproduce the results in this work. Following release of the paper, we will release both a code repository containing the source code for the DARK models as well as the associated scoring functions and analysis scripts. The training data and model parameters will be made available from our servers due to GitHub file size limits.

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

APPENDIX

## A  IMPLEMENTATION AND TRAINING DETAILS

### A.1  ORACLE MODEL AND BACKGROUND MODEL

We use the DMPfold2 model for the $p_\beta(y|x)$ oracle (Kandathil et al., 2020). We received the trained model as well as the training and validation data described in Kandathil et al. (2020) via correspondence with the authors. We note that the code and pre-print for a more recent but non-probabilistic version appears to be available[5], but we refer here to the CASP14 version of DMPfold2 described in the first pre-print (Kandathil et al., 2020), which outputs distograms.

DMPfold2 was chosen as an oracle as it reports favourable performance and, importantly, its ability to perform fast inference directly from an input alignment, or in our case, a single sequence. It uses a combination of two GRUs (Cho et al., 2014) followed by a residual network to predict distograms from a multiple sequence alignment. The input is of shape $[A, L, 22]$ where the $A$ dimension is the number of sequences in the alignment. First, a stack of unidirectional GRUs reduces $A$ to a fixed size, resulting in an output array sized $[L, 512]$, that is fed as input to a stack of bidirectional GRUs that produces the final representation of the input alignment. This representation is then fed to a convolutional ResNet, which produces the final outputs (see below).

We refactored the code for the DMPfold2 model in order to perform batched inference on individual sequences. We find that the IG-score for a single sequence ($L = 100$) can be calculated in approximately 0.06s. For a batch of 100 sequences this reduces to 0.01s per sequence.

**Distogram outputs from DMPfold2**  From a single input sequence of length $L$, DMPfold2 produces an output tensor of size $[L, L, 104]$. The 104D third axis is split into four separate discrete distributions which are individually softmaxed, being sized $[L, L, 2]$ and three of $[L, L, 34]$. The output tensor contains predictions of residue-residue hydrogen bonding, residue-residue $\beta$-carbon ($C\beta$) distance distributions (distograms), and backbone $\phi$ and $\psi$ torsion angle distributions. See Kandathil et al. (2020) for further details.

**DMPfold2 Background Model**  To implement the IG-score we train a background network, $p_\epsilon(y_d)$, as per Anishchenko et al. (2020), using the DMPfold2 architecture and author-provided training data. This model is distinguished with the subscript $\epsilon$. Training the background model constituted replacing the input sequences during training with 64D Gaussian noise, and altering the input layer accordingly, but otherwise following the training procedure described in the DMPfold2 paper. We pre-calculate the background values for a given sequence length by feeding noise to the network 100 times and taking the average of the predictions, leaving a final tensor of size $[L, L, 104]$ containing the resulting predictions. Although this can be calculated on the fly, for simplicity we pre-calculate and save them once. As we focus on $L = 100$, this amounts to one saved array of size $[100, 100, 104]$.

**Using DMPfold2 to calculate the IG-score**  The IG-score is calculated by first predicting the output tensor for a sequence with DMPfold2. Next, $D_{\mathrm{KL}}$ is calculated between each of the four distograms in the tensor and their corresponding pre-calculated background versions. This results in four $[L, L, 1]$ tensors, one for each distogram, containing the scalar $D_{\mathrm{KL}}$ values. For each tensor, the mean $D_{\mathrm{KL}}$ is taken across both dimensions, resulting in four scalars. These four $D_{\mathrm{KL}}$ values are added together with equal weighting. Finally, the IG-score is this value minus the $D_{\mathrm{KL}}$ between the unigram frequency distribution in the predicted sequence and the corresponding distribution in high resolution protein structures from the PDB. This results in a single scalar $D_{\mathrm{KL}}$. The latter values are sourced from the publicly available trDesign code repository[6].

---

[5]https://www.github.com/psipred/DMPfold2
[6]https://github.com/gjoni/trDesign

## A.2    DARK$_0$-*Grad*: Sampling sequences from DMPfold2 with gradient descent

We wished to investigate whether the seed sequences we use in DARK could be generated by a minimizing the negative IG-loss predicted by DMPfold2 with respect to the input sequence. We do so implementing and testing an approach similar to those shown by Killoran et al. (2017) and Norn et al. (2021), but instead using DMPfold2. We begin with an input of shape $[B, L, 22]$ where the first dimension is the number of sequences in the minibatch. To keep compute time results comparable to the other methods presented here we use $B = 100$. The input itself is a one-hot encoded random sequence of $L = 100$. The version of DMPfold2 we use is altered to predict distograms for batches of sequences under the constraint that the alignment is always of size 1, which is not a constraint for us here.

To begin optimization, we compute the distograms from the batch of input sequences, backpropagate the negative IG-score as our loss, and then update the original input $[B, L, 22]$ vector. This is one step. As we are back propagating through a discrete node, and the updated input vector is no longer discrete, we do the following in all optimization steps to make the operation differentiable. We softmax the updated input vector (we refer to this now as a *profile*) in the final dimension and then we sample sequences from the softmaxed residue distributions using a straight-through estimator (Bengio et al., 2013).

We perform 200 of the above steps as we found it sufficient for the negative IG-scores to converge. Instead of sampling final sequences from the final profiles, we track and keep the sampled sequence that achieves the highest IG-score during the 200 steps. We explored a variety of optimizers, profile regularization and normalization approaches, and other variants with the aim of improving the average final IG-score. Despite this, we generally found Adam (Kingma & Ba, 2014) with a learning rate of 1, default $\beta$ parameters, and a temperature of 1.0 on the softmax to be the most consistently effective approach. To calculate the average IG-score of this method, and the average sample time, we generate 1000 sequences from the described approach. These resulting metrics can be seen in Table 1.

## A.3    DARK$_1$-*Adversarial*:

As part of our investigations we were motivated to assess the effect of incorporating a structure predictor directly into the training of our autoregressive DARK models. There has been a range of recent work (Larsen et al., 2016; Esser et al., 2021) exploring the combination of learned and fixed discriminators, like the learned descriminator in generative adversarial networks (GANs) (Goodfellow et al., 2014), with generative autoencoding models like variational autoencoders (VAEs) (Kingma & Welling, 2013; Rezende et al., 2014), vector quantized VAEs (VQ-VAEs) (Oord et al., 2017), and many others. Although this topic has mostly been explore with latent variable models, we wishes to investigate whether such an approach would benefit our autoregressive approach. We note however that there are potentially problems with this approach which we discuss below.

To explore this model, we began with the full training procedure and architecture of the DARK$_1$ model. We then incorporated DMPfold2 by, during a forward pass in training, taking the output probabilities for the batch in question and sampling one-hot sequences combined with a straight-through estimator like in Appendix A.2. The negative IG-score is then calculated as a loss, $\mathcal{L}_{IG}$. We see using DMPfold2 in this way as a kind of adversarial regularizer, thus we termed it DARK$_1$-*Adversarial*. It must be noted that the DMPfold2 parameters are not being updated in this procedure, the model is effectively frozen. We calculate the final loss $\mathcal{L}_{final}$ as,

$$\mathcal{L}_{final} = \mathcal{L}_{AR} + \gamma\mathcal{L}_{IG} \tag{3}$$

where $\mathcal{L}_{AR}$ is the original autoregressive cross-entropy loss and $\gamma$ is a tunable hyperparameter. We performed a small search of $\gamma$ values and found $\gamma = 0.5$ to perform well however this was not rigorously optimized. As can be seen in Table 1, the approach does not perform significantly differently from the original DARK$_1$ model. We in part suspect that this is due to the DMPfold2 model simply reinforcing the loss signal from the cross-entropy against the original sequence; the original sequence is already optimized to have a high IG-score. A less charitable view is that the gradient returning to the decoder model from DMPfold2 is simply contributing noise. As mentioned, there is also a potential issue with the method itself, being that by backpropagating through DMPfold2, which sees the full sampled sequence, we are potentially exposing the decoder to gradient information from the full sequence, breaking the autoregressive nature of the model. Given DMPfold2

is frozen, and we did not see model collapse during training, we do not believe this is occurring, however, we highlight it as necessitating further empirical investigation.

## A.4 AUTOREGRESSIVE MODELS

We build the autoregressive neural network models used in this work with PyTorch (Paszke et al., 2019). We also use several utilities from the NLTK (Bird et al., 2009) package. For all neural network architectures we do mini-batch gradient descent against negative log likelihood (NLL) loss. We use the Adam optimizer (Kingma & Ba, 2014) with default $\beta$ values of 0.9 & 0.999. Early stopping is performed using the validation set. A batch size of 64 is used with a learning rate of $1 \times 10^{-3}$ for the RNN baseline, and $1 \times 10^{-5}$ for self-attention models. The learning rate was not changed according to a schedule as it was not found to provide significant benefit, and so was left static. All training was done with automatic mixed precision as it brought a significant improvement in computational efficiency. Below we describe the architectures for the best RNN baseline model and for the self-attention models.

**Inputs**  For all language models trained on synthetic data, the input is a batch of $B$ sequences in an array sized $[B, L]$. This contains 22 integers corresponding to 20 amino acids, a start token, and a stop token. The length $L$ is the length of the protein sequence with a start token placed at the beginning of the sequence. As is standard for autoregressive models, the target sequence used for calculating the loss is the protein sequence with a stop token placed at the end. All neural networks use embedding layers that use these as integer indices for embedding vectors learned during training.

**RNN architecture**  The best RNN model uses an embedding layer, learned during training, of size 64. The embedded sequence of size $[B, L, 64]$ is fed into an LSTM model with a hidden size $H$ of 128 and 1 layer (Hochreiter & Schmidhuber, 1997). The hidden vector output at each time step of the LSTM is then fed to a linear layer ($H = 128$) $\rightarrow$ ReLU activation $\rightarrow$ linear layer ($H = 22$). The resulting final array, sized $[B, L, 22]$, is then log-softmaxed.

**Xfmr architecture**  The Big and Small Xfmr models use a common architecture but with different numbers of parameters. We describe the architecture using the sizes of the Big model first. An embedding layer of size 384 is concatenated with a positional encoding of size 384 (Vaswani et al., 2017), resulting in an array of size $[B, L, 784]$. The positional encoding uses a maximum length of 101. In comparison, the Small model uses an embedding size of 64, so the input to the following decoder has 128 dimensions in the final axis. We use a standard decoder architecture for autoregressive multi-headed self-attention with causal masking (Vaswani et al., 2017; Radford et al., 2019). This includes 10% Dropout (Srivastava et al., 2014) and the use of Layer Normalization (Ba et al., 2016). We use the Fast Transformers library (Katharopoulos et al., 2020) for implementing models. The Big model uses 12 layers with $H = 784$ and 12 heads, and the Small model uses 4 layers with $H = 128$ and 4 heads. For both models the output of the decoder goes into a linear layer ($H = 128$) $\rightarrow$ ReLU activation $\rightarrow$ linear layer ($H = 22$). The output is produced is also a $[B, L, 22]$ array that is log-softmaxed.

For both RNN language models and self-attention language models, additional decoding strategies beyond simple autoregressive sampling like Top-K sampling (Welleck et al., 2019) were not found to significantly improve results. We mask out cysteine when sampling from DARK models, however we include it in the sizing of the input embeddings and output size to maintain consistency with other approaches. The output array of size 22 accounts for the 20 amino acids, a start, and a stop token. Further investigation of more complex decoding schemes, as well as the use of different positional encoding methods and types of attention, are potential avenues for future research.

## A.5 TRAINING AN UNCONDITIONAL MODEL WITH NATURAL SEQUENCES

Here we describe further architecture and training details of the 'UF50' model used to benchmark the DARK models produced in this work. This model provides a direct comparison of training with natural sequences and training with synthetic proteins from DARK. UF50 is an unconditional language model that uses the same training procedure and architecture as the DARK$_3$ model, with some modifications, and trained on sequences from nature. Specifically, the model is trained on the representative sequences for each cluster in the UniRef50 database (The UniProt Consortium,

2021), yielding a training set of 50,106,395 sequences. UF50 provides a benchmark for an approach similar to the ProGEN model (Madani et al., 2020), if ProGEN was an unconditional model. ProGEN is conditioned on a variety of evolutionary and functional information for the task of protein engineering, making it not directly comparable to the work performed here.

We now describe the modifications used in the UF50 model. During training, missing amino acids and non-canonical amino acids are mapped to an additional 'UNK' token which does not contribute to the loss. Thus, the final vocabulary size is 24. Batches are dynamically padded to the length of the longest sequence, and padded positions are masked for the loss calculation. To reduce the GPU memory required for long sequences, any sequences longer than 100 amino acids (start/stop not included) are clipped to 100. For the target sequence, instead of placing a stop token at position 101, the amino acid sequence in position 101 from the original non-clipped sequence is used. In this way, due to the causal masking, this is exactly the same information available to the model as if it was not using clipped sequences, within the range of 101 amino acids. As all evaluated sequences are of length 100, the clipping has no effect on any conclusions drawn. To provide a level playing field, UF50 uses the same validation set as DARK models. We also use rejection sampling to sample 1000 sequences of length 100 for evaluation and comparison. Additional decoding strategies like Top-K sampling and placing a temperature factor on the next-token softmax were explored but not found to aid performance.

## A.6 FURTHER COMPUTATIONAL EFFICIENCY AND HARDWARE DETAILS

We report all timing estimations, unless stated otherwise, based on using an RTX 2080 Ti graphics card on the same single machine. We report all sample speed timings in seconds per example. Sample timing estimations were calculated with batches of 100 sequences, and calculated on the same single machine. The only example were this was not the case was for $DARK_0$-*Grad* which was performed in batches of 10 due to memory constraints. Model training and sample refinement was performed with resources available at the time but these typically consisted of V100 graphics cards. As such, we report timing for training, refinement, and simulated annealing experiments as they would take on V100 graphics cards. The average time taken for refinement is 51.3 seconds.

## A.7 EXCLUSION OF CYSTEINE

Cysteines are a special case with regards to the other amino acids as they are primarily present in proteins to form disulfide bonds. Disulfide bonds are very strong bonds that form between two cysteine residues. They are, for the most part, present only in proteins that are secreted by the cell or exist on the outward facing surface of the cell membrane. This is due to the fact that the disulfide bond cannot form in the reducing environment of the cytosol of cells. We note that for all the statements, as always with biology, there are exceptions to the rules. In de novo design studies, the inclusion of cysteine is understandably usually framed as the inclusion of disulfide bonds. Marcos et al. (2018) and Bhardwaj et al. (2016) provide recent examples of *de novo* designing with the intent to use disulfide bonds or simply explore if their inclusion provides benefit. Disulfide bonds can be very useful for adding stability to a protein but complicate protein expression and depending on the aim of the design study, it may be disadvantageous or advantageous. From a structural perspective, a disulfide bond can very much 'make or break' a candidate so it requires consideration. Its for this reason that the inclusion of cysteine and disulfide bonds is typically addressed in a methodologically specific way, Bhardwaj et al. (2016) being a direct example of that. This is relevant in our case because it clearly indicates how the default stance is generally to not include cysteine and disulfide bonds unless there is a reason to. Ultimately, its exclusion does not make the problems we tackle in this work any easier and if it were to be included it would be unlikely to add any significant value.

## B    SEQUENCE AND STRUCTURE ANALYSIS

### B.1    VALIDATING THE VALIDATION AND TEST SET

Ideally each seed sample from the black-box oracle could be viewed as identical and independently distributed (i.i.d.). However, given the sometimes unexpected pathologies (e.g. adversarial examples) that can be present in neural networks, we directly measure the diversity of the training set

sequences as well as look for any direct overlap of the validation and test sets with the training set. As can be seen in Table 4, the training set sequences begin to form clusters at around 30% sequence identity.

As a method of comparing generative models between architectures and iterations of DARK, a test set and a validation set of 950 samples each are generated, using the simulated annealing approach of trDesign. We check that the two sets do not share extremely similar or identical sequence samples with the seed set, producing artificially high performance metrics. As such, validation and test set sequences are separately compared to all 15,000 sequences in the seed sample set. This is done using the MMseqs2 (Steinegger & Söding, 2017) search tool with a sensitivity parameter of 7 and one iteration. Given the clustering results of the seed samples (Table 4), we expect a high proportion of the sequences to share some small similarity to sequences in the seed sample set. This was found to be the case; from the validation and test set, 854 out of the 950 sequences in each set were found to have some similarity to sequences in the training set. The mean sequence identity between the similar sequences for both sets was low, being 31% for both. The highest sequence identity found in the validation set was 68%, and for the test set 60%. These results are as would be expected; they suggest that both sets are sampled from the same distribution as the seed set while not containing any near-identical copies. This is acceptable, given that this is an unsupervised task using synthetic samples, where the aim is to learn that distribution. We make the distinction as had natural sequences been used a more stringent test and validation splitting based on structure and evolutionary relationships would be required.

As part of investigating the effects of DARK iterations, we also performed this search process with the test set for the 500,000 examples of the third iteration's training set. The number of examples in the test set that had any detected hits increased a small amount, from 854 ($N = 1$) to 922. The mean sequence identity of the hits increased slightly from 31% to 33% and the highest identity hit increased from 60% to 66%. Again, these values are acceptable, and even potentially a positive sign. One likely cause for the increase in the performance of models across iterations of DARK is that the area of sequence space covered by the training set is progressively growing in an unsupervised manner. Even from a pessimistic perspective, if the $N = 3$ model has better likelihood estimates on the test set because the previous iterations generated training samples closer in space to the test set, then it suggests that the iterations are indeed expanding and filling in unseen parts of the distribution.

## B.2 THE STRICT STRUCTURE-BASED TEST SET AND DARK$_2$-*STRICT*

We take two key approaches to providing a stringent measure of how well DARK generalizes. Both are enabled by first using AlphaFold to generate all-atom structure predictions, and pLDDT scores, for all 100,000 sequences contained in the DARK$_2$ training set. This set covers the 15,000 initial seed sequences and the 85,000 sequences from the first iteration of DARK. We also assign each structure a topology in the form of a *topology string*. This describes the topology by the secondary structure elements going from the N-terminus to C-terminus. For example, if this had been done for the de novo designed 'Top7' protein (PDB ID: 1QYS) the topology string would be '$\beta\beta\alpha\beta\alpha\beta\beta$' or 'bbababb'. We also repeat this procedure for the 950 sequence validation set.

The first way we use this 100K set is to group sequences by their topologies and show that the first iteration of DARK generates sequences with structures unseen in the initial 15K seed set. The seed set contains 421 unique topologies and the 85K set contains 1011 unique topologies, with an intersection of 373 between the two. As such, there are 638 topologies in the 85K set that were unseen in the training set. This accounts for 1476 sequences, of which 225 sequences (15.2%) have a pLDDT$\geq$ 70. This provides strong evidence that the DARK iterations are expanding the training sets to include sequences with unseen and, importantly, ordered structures.

The second way we use this set is to construct, and train a model on, a new training and test set split based on structure, as is considered the gold-standard in evaluating structure prediction methods (Chothia & Lesk, 1986; Söding & Remmert, 2011). We refer to the model trained and tested on this new split as DARK$_2$-*STRICT*. We construct the test set by first randomly sampling 101 topologies, approximately 10%, from the 1059 unique topologies in the 100K set, ignoring the 100 most common topologies and uninteresting topologies, being a single helix (131 sequences) and only coil (3 sequences). This 101 topologies accounts for 413 sequences which are removed from the 100K set, and become the *strict* sequence test set (See Appendix B.2 for the test topologies and further

details). We also remove and discard 4 sequences from the 950 example validation set that have a strict test set topology. The original 950 example test set is not used here.

Next we use the MMseqs2 search tool (Steinegger & Söding, 2017) with the maximum sensitivity parameter (-s 8.5) to aggressively search for potential similarity between the strict test set sequences and sequences in the remaining 99587 training examples and 946 validation examples. We then discard any training and validation sequences that MMseqs2 returns as hit, meaning it is potentially similar to a test set sequence. This reduces the size of the training and validation sets to 87551 and 621 examples respectively. We then use the 87551 sequence training set, 621 sequence validation set, and 413 sequence strict test set to train and evaluate the DARK$_2$-*STRICT* model.

We note that this approach to assigning structures to topologies is an aggressive one as it does not consider the different packing arrangements that the secondary structure elements can have. Thus, our measures of the number of topologies present in the set is likely a lower bound on the number of topologies that would be assigned by a more intricate approach like that done for CATH (Orengo et al., 1997). This may make the test set we have constructed more difficult however we wish to take a conservative approach and challenge the model, so we find this acceptable.

**The 101 randomly selected topologies, and their sequence counts, making up the STRICT test set** The 'a' is for alpha helix and 'b' is for beta sheet. The 101 topologies cover a total of 413 sequences:

*bbabbbabbbb (7), abaaaab (2), baaababab (1), aaabbbabb (1), babb (3), bbbbbbbbbbbab (6), bababaaa (1), abbaabbbb (2), bbbabbababb (3), bbabbbaabb (3), bbabbbabababbab (1), bbbbb-bababb (3), aaabbb (5), babbabbbba (2), abbbbbbbabbab (1), aabbaa (32), bbaababa (1), bbbbbb-baab (4), bbabbbabbab (1), abaaabbaa (1), abbaaabbaa (1), bbabbabbbbbabbb (1), abbaabba (7), babbbabab (3), babbabbbabb (19), bbbbabbaabba (1), aabaaabbaa (1), babbbbbabbbb (2), bb-baabbba (2), babababbbbb (2), bbabbabbababb (6), babbbaab (2), bbbababbb (18), babbbabba (3), bbbaabaa (1), babaabb (14), abaabbabbab (1), bbababbbb (19), baaaab (9), bbaaaabb (1), abaabaabaabaa (1), bbaabab (4), abbbabbabbbbb (1), bbbbaabbbb (1), bbbbbbbbaab (1), baab-baab (2), bbababababab (1), bbabbabbabbab (2), aabbbab (4), bbaabbabbb (1), bbbbbbbababb (5), bbaababaab (1), bbabbbbab (29), abbbbabb (22), aaaabab (2), babbbbbabbb (4), bababbabb (6), babaabbbaab (1), bbabbabbbbbbb (1), bbbbabbbbabbb (2), babbabbabab (1), bbabaabbb (2), baabbbbbb (5), aaabaabaab (1), bbbaabb (7), abbbbabbabb (1), bbaabbbbaab (1), ababaaa (1), babaabab (7), bbabbbbaa (3), aabbaabb (4), abbabababb (5), abbbabbabbb (1), bbaabaabb (1), bbb-baab (7), baababb (8), aaabaabba (1), ababababaa (1), bbbabbbbbbabbab (1), bbabbbbabbbb (4), abbaaaaa (3), abbbbababb (1), bbbbabbbaab (1), abaabbabba (1), bbbbabbbbabbb (2), bbabbb-babbbabb (1), babbbbba (26), aaabbbabb (1), abbbabbbbbbb (1), babbbbabab (3), bbaabbabbba (1), bbbbbabbabbb (1), abbabbabbb (2), bbbbaabbbabba (1), bbabbbbbbbba (1), babbababb (8), abb (5), bbaaabbbbba (1), bababaab (2), bbbbabbbbbbbbb (2), babbbbabba (3)*

B.3 PREDICTING STRUCTURE WITH ALPHAFOLD

We use the publicly available version of AlphaFold[7] for predicting structures and as part of a quality metric. In both cases, we use the standard AlphaFold inference pipeline, meaning we predict with the five 'CASP14' models, take the predicted structure with the highest pLDDT as the final result, and perform relaxation. We use adapted inference scripts to accommodate for predicting with only a single sequence and without using the template pipeline or the multiple sequence alignment (MSA) pipeline. This consisted of setting all models to use the same default configuration, as they all use variants on the default, and then setting all to not use templates. The MSA clusters are reduced to 1 sequence, being the single input sequence, and the deterministic flag is set to true. The number of recycles is set to 4. Following structure prediction, the standard relaxation is performed for the best final structure. We note that relaxation is not necessary to produce a final structure, thus it can be said that a trained language model, like those we present, with AlphaFold provides a direct way to sample from a joint distribution of sequence and structure, and without additional conditioning information. Predicting structure with AlphaFold provides the pLDDT scores at each

---

[7]https://github.com/deepmind/alphafold

residue position. We take the average of these for the pLDDT score of a given sequence. Structures are visualized with ChimeraX (Pettersen et al., 2021).

### B.4 ALPHAFOLD PREDICTION CONFIDENCE AND DARK$_3$ FOR RECENTLY RELEASED DE NOVO DESIGNED STRUCTURES IN THE PDB

As we use the pLDDT from AlphaFold as a metric for measuring structural order, we wish to double-check that AlphaFold assigns good or better pLDDT scores to *de novo* designed sequences in the PDB that were not used as a part of its training set. We manually curate a list of 26 of these sequences from the PDB. The PDB IDs containing these are 6MRR, 6MRS, 6NUK, 6MSP, 6XWI, 6XXV, 6VTW, 7KBQ, 6D0T, 7JH6, 6XEH, 6X9Z, 6WVS, 6E5C, 6DS9, 5TX8, 5W9F, 6YWD, 6Z35, 6W6X, 6W90, 6WI5, 6W3W, 6VGB, 6W70, 6YQY.

This is not a comprehensive list as, amongst other reasons, some sequences were not included if they were very sequence-similar to an already selected structure or if they are a large tandem repeat structure (e.g. 'armadillo repeats') to avoid biasing the average. This is still a biased set of structures as they are the product of whichever goal a given study was investigating, but the pLDDT scores are still informative. We note that although the PDB has a 'DE NOVO PROTEIN' classification, this includes engineered, designed, and *de novo* designed proteins. The average calculated pLDDT of all sequences in the list is 88.1 ($\sigma = 4.41$), a good score, as is expected.

We also calculate the likelihood of these sequences under DARK$_3$ compared to random sequences of length 100. We ignore the 5 sequences that contain cysteine. To account for DARK having been trained on static length proteins we ignore the prediction of the 'stop token' and ignore all residues in a sequence above position 100 if they exist. We calculate the bits per character (BPC), where characters are residues, across 100 random sequences and the 21 designed sequences. The random sequences have a mean BPC of 5.88 bits and the designed sequences have a mean BPC of 4.96 bits. In having a lower BPC suggests that the DARK models have learned a distribution of sequences that incorporates structure and is capable of producing *de novo* designed sequences more so than random. It is also unlikely that by random chance the DARK models have generated and learned from sequences in the same parts of the dark sequence space

### B.5 BENCHMARKING WEAKLY CONDITIONED METHODS

We provide results for benchmarking with Good+ pLDDT proportions on two recent weakly conditioned generative protein design models, which we refer to as gramVAE (Greener et al., 2018) and gcWGAN (Karimi et al., 2020). These both use a fold level description that we call weak as they are far removed from the explicit structural information provided in fixed backbone models (Ingraham et al., 2019; Strokach et al., 2020; Cao et al., 2021). We stress that these models are not directly comparable to the unconditional models explored in this work. Furthermore, we provide these only to gain a small degree of insight but this is significantly muddied by both models being able to 'cheat'. If either of them reproduce a sequence from their training set, then it is extremely likely that the same sequence and its structure are in the AlphaFold training set[8] and so likely to be assigned a higher pLDDT than would otherwise be expected. Nonetheless, we provide these benchmarks as they provide some small degree of context. Even with extra information from conditioning variables and this ability to cheat, they are still outperformed by DARK$_3$ (Table 8). We speculative that this difference can even be seen visually in the predicted structures shown by both models (Figure 5c of Karimi et al. (2020) and Figure 6 of Greener et al. (2018)) which contain clear disordered regions.

The gramVAE[9] uses a context-free grammar of SCOP protein folds (Andreeva et al., 2020) as a conditioning variable for a Variational AutoEncoder (VAE). Both encoder and decoder use non-linear layers and sequences are padded to a maximum length of 140. The gcWGAN is a conditional Wasserstein Generative Adversarial Network that also pads length to use fixed-sized networks (Karimi et al., 2020). It conditions on a small latent representation of folds, learned from performing kernel PCA on an all-vs-all distance matrix between all folds in the dataset, where the kernel used is symmetric TM-score (Zhang & Skolnick, 2005) between two folds. The publicly available

---

[8]AlphaFold makes the blanket statement that all structures from the PDB with a maximum release date of 30 April 2018 are used in its training.

[9]https://github.com/psipred/protein-vae

implementation[10] provides 600 sequences generated for the 6 folds (100 sequence each) in its test set using the gcWGAN and the gramVAE in a head to head comparison. Being publicly available and a direct comparison, we use these provided sequences to calculate Good+ pLDDT proportions for both models. The gramVAE achieves $21.0\%$ and the gcWGAN achieves $2.83\%$.

We note that the majority of the folds in the gcWGAN test set are in the gramVAE training set, which likely explains part of the performance difference. We are not concerned by this as the DARK models still outperform the gramVAE with an artificial boost. We also note that the gcWGAN model does not discretize the output of the generator and use any approaches to backpropagating through the discrete output variable (Bengio et al., 2013; Jang et al., 2016). This provides the discriminator a trivial learning scheme: place the decision boundaries at the edge of the probability simplex of tokens at any position. This may explain part of the gcWGAN's poor performance. In both cases, we recognize that these models are not ideally suited for sequence problems and have reported poor performance elsewhere (Cao et al., 2021). We expect future weak conditioning methods will provide more competitive benchmarks. Ideally, we hope that future investigations into novel unconditional models will provide direct benchmarks, establishing the best approach to unconditional modelling.

## B.6    DARK MODEL SAMPLES ARE DISTINCT FROM NATURAL PROTEINS

We find that samples from the $DARK_3$ model are distinct from sequences in nature. A sequence search against UniRef100 of the 1000 evaluated sequence samples (without refinement) returned no hits for all sequences except one, which had 5 weak hits. Two pairs of these 5 sequences were extremely sequence-similar. Repeating the sequence search with a different tool, HHblits (Remmert et al., 2012), no natural sequences were found that met the HHblits default threshold value (an E-value below $1 \times 10^{-3}$).

## B.7    SEQUENCE SEARCHING AND CLUSTERING

We use the MMseqs2 package (Steinegger & Söding, 2017) to perform sequence searches and sequence clustering. For clustering, the default clustering pipeline and settings are used with sequence identity as the threshold. If not specified, this is assumed to be a threshold of 30% sequence identity. For sequence searching we use the default pipeline of MMseqs2 and a high sensitivity parameter of 7.0 with 1 iteration against UniRef100 (03-2021 release) (The UniProt Consortium, 2021). Any sequence searching done with HHblits (Remmert et al., 2012) uses an E-value cut-off of $1 \times 10^{-3}$ with 1 iteration against UniRef30 (06-2021 release) (Mirdita et al., 2017). In this work, we call any two sequences similar if MMseqs2 (or HHblits) returns a hit between them, meaning it has detected similarity. Sequences that have a, for example, 40% sequence identity or above are also defacto assumed as similar.

## B.8    STRUCTURE CLUSTERING

To cluster predicted protein structures we perform single-linkage agglomerative clustering using TM-score (Zhang & Skolnick, 2005) as a similarity metric, meaning an all-against-all matrix of TM-scores is calculated for each structure in the set of 1000 samples. We use a TM-score of 0.4 as the cutoff for selecting best cluster centroids and a TM-score of 0.65 for single-linkage clustering around the centroids.

## B.9    GENERATING DE NOVO DESIGNS WITH DARK AND ALPHAFOLD REFINEMENT

**Background**    To give a concrete demonstration of how $DARK_3$, and models like it, can be used in developing novel methods, we develop a simple but novel design method. The method we develop, termed AlphaFold refinement, tackles what could be considered the next step for using a model like DARK. Although we have used AlphaFold primarily to evaluate samples, and by extension models, it can also be used to improve candidates. An obvious first step is that AlphaFold produces both distogram based predictions and an all-atom structure prediction, so a simulated annealing approach like trDesign's (Anishchenko et al., 2020) could be used to sample sequences. We note that the

---

[10]https://github.com/Shen-Lab/gcWGAN

all-atom prediction can be considered a point estimate of the distogram, but the all-atom model is ultimately the output of the most interest to us for design.

**The challenges of a simulated annealing approach with AlphaFold**  Unfortunately, this approach poses two problems. The first is that, at its fastest, meaning running AlphaFold with no recycling (Jumper et al., 2021), each forward pass takes approximately 5 seconds. This is in contrast to 0.064 seconds with DMPfold2. Performing a simulated annealing run with the same schedule as we use would take ∼2.3 days which is prohibitively long. The second problem is one we find in extending the approach itself. Directly optimizing on the AlphaFold IG-score (AIG-score) itself appears to produce, what amounts to adversarial attacks on AlphaFold. This can be seen in Figure 2A & 4, and is particularly evident in the low AIG-scores (low compared to our approach) but extremely high pLDDT scores. The visual confirmation of this is evident in Figure 4 where all structures produced are a very unrealistic single long helix. We implement the AIG-score as the IG-score but using AlphaFold to calculate the distogram prediction from which the $p(y|x)$ term is calculated. As we do not have the resources that were use to train AlphaFold (Jumper et al., 2021), we calculate a proxy background distribution by predicting the structure of 1000 random sequences of $L = 100$ and then averaging their resulting distogram predictions together into one vector.

**Using DARK$_3$ to enable AlphaFold design**  Introducing DARK$_3$ into this problem brings several benefits. First and foremost it allows us to generate diverse and high confidence samples from AlphaFold without them being adversarial attacks. It also reduces the number of design steps such an approach would take, and adds a measure of control into the design process. We call this method AlphaFold refinement as it is similar in concept to the refinement done with DMPfold2 between iterations of DARK. DARK$_3$ allows us to generate large sequence databases, with predicted structures, that have a diverse range of topologies. These include structures with topologies that have yet to be successfully de novo designed, and many with high confidence stable structures that can be used as *de novo* scaffolds into which functional sites are grafted (Woolfson, 2021). In short, DARK$_3$ produces sequences that have structures that are of design interest.

**AlphaFold refinement**  AlphaFold refinement begins with a sequence sample from DARK and its AlphaFold predicted structure, chosen by the designer. This is how control is introduced into the design process. For a graphic displaying the full AlphaFold refinement procedure see Figure 2. The next step is to take the predicted all-atom structure and calculate its one-hot encoded distogram representation (Jumper et al., 2021). We then bootstrap this representation as a fixed optimization target. The actual optimization that occurs is a greedy hill-climb on the sequence against the Cross-Entropy between the predicted distogram and the bootstrapped target. As we use all five parameter sets when we use AlphaFold, we treat the distogram prediction at each step as if it were an ensemble of five AlphaFolds; we average the five predicted distograms as one final predicted distogram with which to calculate the Cross-entropy. We run this for AlphaFold refinement for 10000 steps, as can be seen in Figure 2C, however we found comparable results could be achieved in 7500 steps, being 10 hours per sequence on a single consumer-grade GPU (RTX 2080 Ti). This is significantly more efficient than an approach like using Rosetta which can take thousands of CPU hours. In summary, we mutate the sequence and keep mutations that minimize the Cross-Entropy to the original structure. The underlying reason for undertaking this approach is that the structure and sequence we are interested in will, more often than not, have a lower pLDDT than is possible with a similar sequence that folds to the same structure. In short, there is always room for improvement, and AlphaFold with DARK$_3$ provides a way to address this.

This provides two benefits. The first is that it forces the objective to keep the original structure we are interested in. We find that this also provides a shorter and, we speculate, more direct optimization path to a high confidence structure. There stands to reason as we giving a fixed objective, a much easier task, rather than letting the random walk find arbitrary minima. This is evident in the average AIG-score and pLDDT score improving vastly while the final sequences remain very similar from a sequence identity perspective. What is particularly notable is that the average pLDDT score of the 10 examples explored is very similar to that of recently released de novo designed structures in the PDB (Figure 2D & 2E). The 10 examples we take are chosen from the 5000 DARK$_3$ samples previously generated in a semi-random we fashion. We chose them randomly but rejected any with the exact same topology, accounting for packing, and any single sheet (CATH 2.20; Orengo et al. (1997)) topologies simply because they are uninteresting. The 10 examples cover 4 $\alpha$ & $\beta$ structures,

3 $\alpha$ structures, and 3 $\beta$ structures. This rather even spread of examples was entirely coincidental but convenient.

Another aspect of using the distogram is that it provides a softer means of keeping the original structure. In essence, by working in a probabilistic space, AlphaFold can perform small rearrangements to the original structure to improve its confidence while not straying to far. This is difficult to quantify but can be seen in Figure 2D Example 8, a complex $\alpha$ & $\beta$ topology where several of the sheets are brought together in the lower half of the structure (as visualized) to improve packing. To summarize, in the limit of AlphaFold's prodigious accuracy, AlphaFold refinement provides a way to refine design candidates of interest and improve the confidence (and order). Another way of viewing this approach is as an entirely neural network driven approach flexible-backbone design, here used as part of an overall *de novo* design method. In that vein, we note that this approach could be extended very easily to other tasks in protein design like fixed-backbone design.

**Designing a novel immunoglobin**    As an example of using this approach we take a immunoglobin from the 5000 DARK$_3$ samples that we previously generated and then AlphaFold refine it. We take the first example of an immunoglobin in the set and find that we successfully refine its structure to a high pLDDT and AIG-score while not altering the original approach. The results of this design can be seen in Figure 3. This provides a concrete example of using DARK$_3$ and AlphaFold to design a high confidence candidate for a topology of high scientific interest that traditional approaches like Rosetta are known to struggle with.

# C    ADDITIONAL FIGURES AND TABLES

Table 3: **Samples from DARK models have diverse sequences and predicted structures.** Sequence and structure diversity is shown with the number of clusters (of 1000 samples) estimated by sequence clustering and structure clustering respectively. Change in the Good+ pLDDT between iterations is also included.

| DARK Model | Good+ pLDDT (%) ↑ | Sequence Clusters ↑ | Structure Clusters ↑ |
|---|---|---|---|
| $DARK_1$ | 28.0 | 1000 | 829 |
| $DARK_2$ | 31.0 | 973 | 835 |
| $DARK_3$ | 32.3 | 962 | 822 |

Table 4: **Diversity of the sequence examples in the training set.** Diversity at each iteration is measured by number of sequence clusters found by clustering with four different sequence identity (Seq. ID) thresholds. The number of training examples (Train Ex.) at each iteration is included.

| Iteration | 99% Seq. ID | 90% Seq. ID | 50% Seq. ID | 30% Seq. ID | Train Ex. |
|---|---|---|---|---|---|
| $N = 1$ | 15,000 | 15,000 | 14,994 | 11,076 | 15,000 |
| $N = 2$ | 100,000 | 100,000 | 99,942 | 66,509 | 100,000 |
| $N = 3$ | 500,000 | 500,000 | 498,077 | 231,425 | 500,000 |

Table 5: Showing baseline results of refining random sequences against the AlphaFold IG-score using a greedy hill-climb. Sequence identity is between a sequence before refining and the final sequence produced by refining. This was ran for 20,000 steps.

| Example | Best AlphaFold IG-score ↑ | Sequence Identity to Starting Sequence ↑ |
|---|---|---|
| 1 | 0.865 | 0.16 |
| 2 | 0.861 | 0.14 |
| 3 | 0.849 | 0.12 |
| 4 | 1.210 | 0.24 |
| 5 | 0.837 | 0.15 |
| 6 | 0.855 | 0.18 |
| 7 | 1.227 | 0.22 |
| 8 | 0.859 | 0.13 |
| 9 | 0.852 | 0.21 |
| 10 | 0.834 | 0.14 |
| **Average** | **0.925** | **0.169** |

Table 6: Showing baseline results of using a trDesign-like approach with AlphaFold. It refines random sequences against the AlphaFold IG-score using simulated annealing. Sequence identity is between a sequence before refining and the final sequence produced by refining. This was ran for 20,000 steps.

| Example | Best AlphaFold IG-score ↑ | Sequence Identity to Starting Sequence ↑ |
|---|---|---|
| 1 | 1.204 | 0.03 |
| 2 | 1.198 | 0.07 |
| 3 | 1.210 | 0.07 |
| 4 | 1.214 | 0.02 |
| 5 | 2.009 | 0.05 |
| 6 | 1.224 | 0.06 |
| 7 | 1.204 | 0.05 |
| 8 | 1.196 | 0.03 |
| 9 | 1.205 | 0.10 |
| 10 | 1.204 | 0.04 |
| **Average** | **1.287** | **0.052** |

Table 7: Showing results of performing AlphaFold refinement with $DARK_3$. It refines $DARK_3$ sequences against the Cross-Entropy between AlphaFold's initial step 0 distogram and the current sequence's distogram, indirectly improving the AlphaFold IG-score, using a greedy hill-climb. Sequence identity is between a sequence before refining and the final sequence produced by refining. This was ran for 10,000 steps as it was sufficient for convergence and so stopped as to not waste resources.

| Example | Best AlphaFold IG-score ↑ | Sequence Identity to Starting Sequence ↑ |
|---|---|---|
| 1 | 2.387 | 0.33 |
| 2 | 2.190 | 0.39 |
| 3 | 1.992 | 0.38 |
| 4 | 2.160 | 0.36 |
| 5 | 2.008 | 0.32 |
| 6 | 2.145 | 0.39 |
| 7 | 2.197 | 0.42 |
| 8 | 1.316 | 0.43 |
| 9 | 2.317 | 0.40 |
| 10 | 2.329 | 0.33 |
| **Average** | **2.104** | **0.375** |

Table 8: Good+ pLDDT scores from AlphaFold (higher is better) are shown for unconditional language models, uniformly random sequences, and two recent conditional models that use non-specific (weak) structure information for conditioning.

| Language Model | Good+ pLDDT (%) ↑ | Conditioning | Training Data |
|---|---|---|---|
| Random | 1.8 | - | - |
| gcWGAN (Karimi et al., 2020) | 2.8 | Structure | Natural |
| gramVAE (Greener et al., 2018) | 21.0 | Structure | Natural |
| UF50 | 15.9 | - | Natural |
| $DARK_1$-*Adversarial* | 28.0 | - | Synthetic |
| $DARK_1$ | 28.0 | - | Synthetic |
| $DARK_2$ | 31.0 | - | Synthetic |
| $DARK_2$-*STRICT* | 29.2 | - | Synthetic |
| $DARK_3$ | **32.3** | - | Synthetic |
| $DARK_3$ (Ref) | **37.3** | - | Synthetic |
| $DARK_0$-*Seed Samples* | 40.8 | - | Synthetic |

**Example: *De Novo* Design of an Immunoglobin Fold
Using DARK3 and AlphaFold Refining**

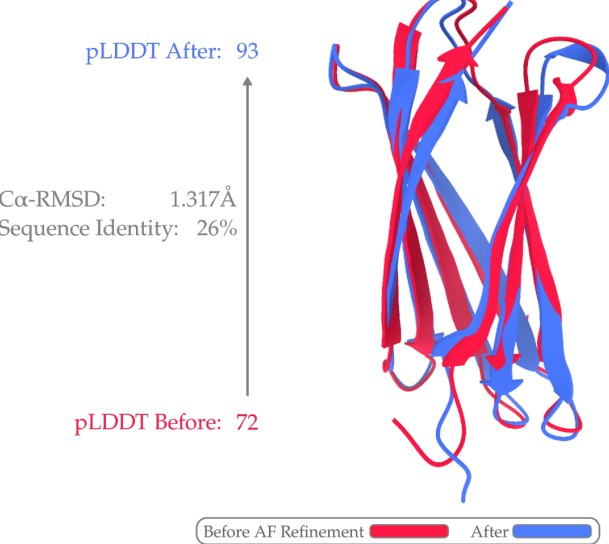

pLDDT After: 93

Cα-RMSD:           1.317Å
Sequence Identity:   26%

pLDDT Before: 72

Before AF Refinement ▬▬  After ▬▬

Figure 3: Showing the structure of a DARK sampled sequence predicted to have an immunoglobin fold by AlphaFold. The structure of the sequence is shown after AlphaFold refinement along with the change in pLDDT, the Cα-RMSD difference between the two structures, and the sequence identity between the two sequences. The structures are visualized with ChimeraX (Pettersen et al., 2021)

**Adversarial Examples Produced by Simulated Annealing of the AlphaFold IG-score**

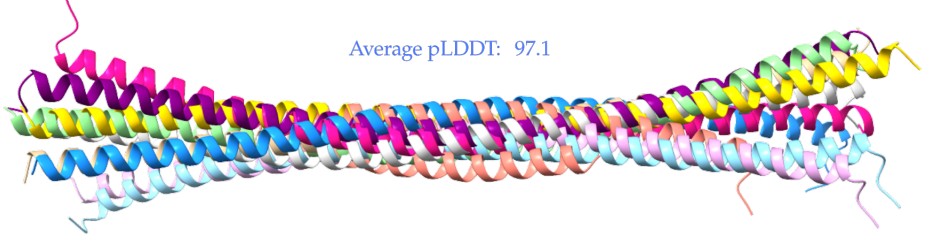

Average pLDDT: 97.1

Figure 4: The superimposed structures of 10 examples produced by simulated annealing of the AlphaFold IG-score, in essence, a trDesign approach with AlphaFold. All 10 examples achieve a low IG-score of 1.28 (our method achieves 2.10) but an extremely high mean pLDDT of 97. This direct approach to maximizing the AlphaFold IG-score effectively generates adversarial examples, evident in all samples being predicted to have the same single long helix. The structures are visualized with ChimeraX (Pettersen et al., 2021)

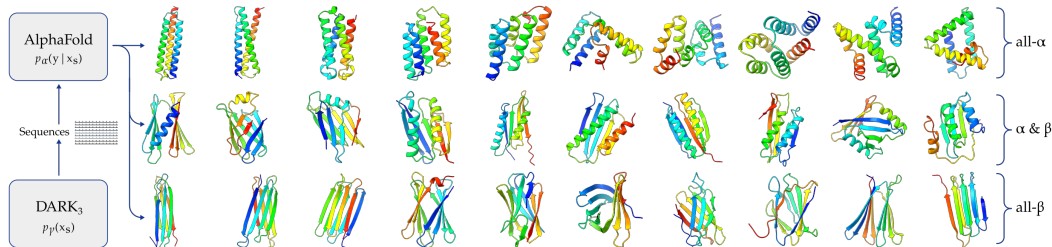

Figure 5: **DARK3 paired with AlphaFold unconditionally samples diverse predicted structures.** We show varied examples of protein structures, predicted by AlphaFold, for sequences sampled directly from DARK3 without any refinement. These structures have a variety of different folds, and are visualized with ChimeraX (Pettersen et al., 2021). For clarity, we split the rows into examples of all-$\alpha$, $\alpha$ & $\beta$, and all-$\beta$ structures.

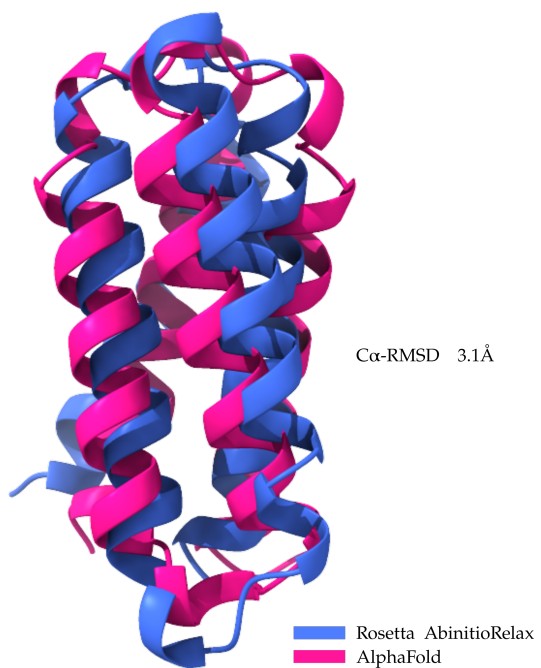

Figure 6: **Agreement between Rosetta and AlphaFold structure prediction.** Predicted structures of a sequence sample (ID 209a) using AlphaFold and Rosetta AbinitioRelax. Both approaches are in agreement, producing close to the same structure. For the latter, the best predicted structure with the lowest Rosetta all-atom score is used, being -279. The root-mean-square deviation between $\alpha$ carbons in the two different predictions is included.

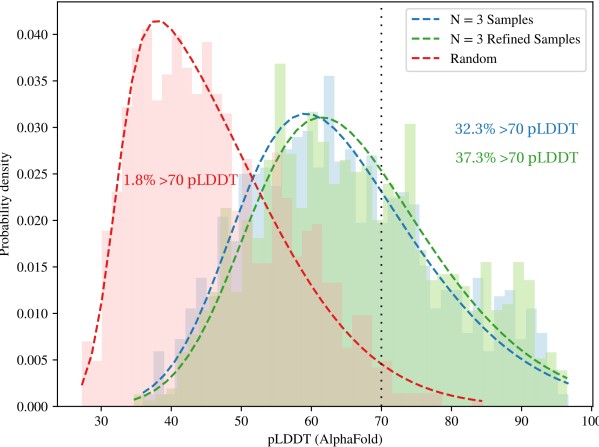

Figure 7: **DARK models generate samples with confidently predicted structures.** A Normalized histograms of the pLDDT scores produced by AlphaFold when predicting structures for three sets of sequence samples, each containing 1000 sequences. A skew-normal is fit to each set and plotted for clarity. *Random* refers to uniformly random sequences of $L = 100$. The Good+ pLDDT (pLDDT$> 70$) proportion is also included. $N = 3$ refers to the DARK$_3$ model.

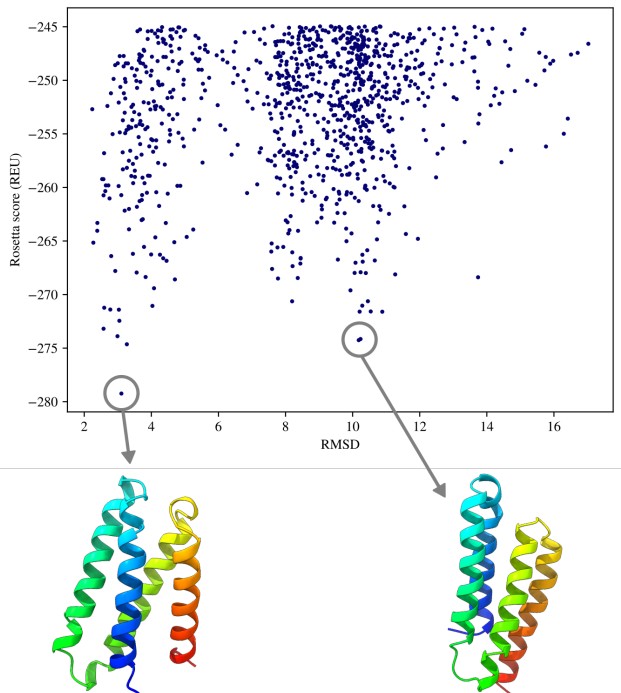

Figure 8: **Rosetta predicts correct fold and potential alternative conformation.** The top 1000 predicted structures (by Rosetta score) for a sampled sequence, of 10000 structures, produced by the Rosetta AbinitioRelax protocol. Rosetta scores are expressed in Rosetta Energy Units (REU; lower values are better). RMSD is root-mean-square deviation (RMSD) in Å from the AlphaFold structure prediction of the sampled sequence using $\alpha$-carbon atoms. The best model is bottom left but a second peak suggests an alternative conformation may be possible (bottom right). We note a similarity to the multi-peak distributions presented in Figures 2B & 2C of Norn et al. (2021).

