# OpenReview forum: "Design in the Dark: Learning Deep Generative Models for De Novo Protein Design"
_ICLR.cc/2022/Conference — ICLR 2022 Submitted_

### Official Review · Reviewer_g4tX · 2021-10-22

**Correctness:** 1
**Technical Novelty And Significance:** 2
**Empirical Novelty And Significance:** 1
**Recommendation:** 3
**Confidence:** 5

**Main Review:**

### Generated sequences are highly likely to be extremely biased toward a specific subset of sequences
The way these sequences are generated is that they start with some seed sequences optimized out of DMPfold2. Then, the sampling model $p_{\gamma^{n}}$ is fit to this distribution, and then used to generate sequences from this distribution. These sequences are then further optimized using the same DMPfold2 model to generate more training examples. Throughout this procedure, $p_{\gamma^{n}}$ is almost certainly going to learn a very similar distribution to the initial seeds, and is unlikely to generate novel examples beyond these seeds which are significantly different in any way. Furthermore, these seeds are generated from a single model (DMPfold2). This already biases these sequences toward the sequences used to train DMPfold2.

Overall, this iterative process (i.e. sampling sequences and maximizing IG-score) doesn't seem like it would help at all with generalizing to novel sequences, and only serves to focus the sequence model on a specific sub-distribution. I expect any improvements in structural likelihood that stems from iteration to just be refining the structural likelihood of the initial seed sequences, which were selected suboptimally (via simulated annealing).

The number of sequence clusters (as a measure of the diversity among generated sequences) was shown for the three DARK iterations, but it is not clear how to interpret numbers of 1000, 973, and 962. Is that big or small? How does that compare to the baselines? Also, having 1000 clusters for 1000 sequences may suggest that this metric is not very reasonable to begin with. With that many sequences, one would expect at least a few clusters of size greater than 1 just by random chance (usually).

Furthermore, the authors claim that the 1000 samples drawn from DARK have no strong similarity to natural proteins in UniProt. This needs to be quantified.

### The model is still conditioned on structure
Even though the paper claims that the DARK method is unconditional, it still depends very heavily on structure. Structural information is coming in from the conditional model DMPfold2, which has an extremely central role in the algorithm. Thus, even though the final model $p_{\gamma^{n}}$ only considers sequences, it has structural information built into it. Specifically, it is _conditioned_ on the initial seed samples and DMPfold2.

### There is no real validation of the generated sequences' structures
Using AlphaFold is unfortunately not a convincing validation of these sequences' structural likelihoods. Although it is technically a different neural network from DMPfold2, it is still only an estimate with no theoretical guarantees. That is to say, it is not "real". Without the ability to synthesize and test these generated sequences for structure experimentally, one could use existing structured proteins and use the sequence model to score these sequences. The model should give structured proteins a higher score than disordered proteins.

### Outperforming natural sequences is concerning
The fact that the model doesn't seem to do so well with natural sequences is very concerning, because natural sequences (excluding those that are disordered) are typically highly thermostable and thus structured. In fact, natural sequences in many ways should be the upper bound of how well the generating model performs (unless the claim is that these generated sequences are _more_ structured than natural sequences).

### Missing some baselines
Previous works have also attempted to generate _de novo_ proteins. For example, Alley, et. al., 2019 (UniRep) can be used to generate sequences that can also be scored by AlphaFold. Anand & Huang 2018 also generate novel proteins that are structured. Notably, UniRep is truly unconditioned on structure (i.e. there is no notion of structure in their model), whereas DARK still conditions on structure. In many ways, DARK lies on the spectrum between these two previous works (in terms of reliance on structure).

It should also be noted that UniRep is capable of generating sequences that are structured, which is shown in the paper.

### Some choices in the algorithm are unclear or of questionable soundness
The choice of the loss function is not quite clear. For example, why use the background model $p_{\epsilon}(y_{d})$? Why not just maximize $p_{\beta}(y_{d}\vert x_{d})$? Additionally, is the sequence prior needed? What happens if it is removed? Is there a weight $\lambda$ for these two halves of the loss function?

It is also not clear why examples are concatenated (step 6 of the algorithm). Do old examples need to be present even though larger and larger sets of sequences are being generated?

The use of a different model architecture for each iteration of DARK is concerning. This suggests that the developed method may be extremely overfit to this specific task/dataset. There doesn't seem to be a justification for the different architectures.

Also why are Cys residues removed? Given that disulfide bridges are a critical part in maintaining tertiary structure in many proteins, a model that does not consider their existence raises some red flags.


**Summary Of The Paper:**

The authors propose a way to generate novel protein sequences that are likely to have folded structures. To do so, they start with a pre-trained sequence-to-structure predictor, and use simulated annealing to generate a set of initial sequences that are likely to have a structure. These seed sequences are used to train an unsupervised sequence model (an autoregressive transformer). More sequences are generated from this sequence model, and they are optimized to maximize the likelihood they have structures (using the pre-trained sequence-to-structure model), and are added to the growing set of training sequences. This process is repeated for several iterations to finetune the unsupervised sequence model. The authors show that sequences generated from this sequence model are likely to be structured as predicted by AlphaFold.


**Summary Of The Review:**

The problem of generating novel protein sequences is a popular one. Although this submission develops a somewhat novel approach to solving this problem, the method's success is dubious, if not absent. The overarching claim that the method is unconditional is not quite true, and the sequences generated are almost certainly highly biased and limited. There is also no real validation of the generated sequences, and minimal comparisons to existing work in the same space. For these major methodological and technical issues, I unfortunately cannot support its acceptance.

---

> ### Author Response · Authors · 2021-11-27
> **Response to Reviewer g4tX**
>
> Thank you for your feedback and comments. Please see our detailed responses to the individual headings below:
>
> ### RE: Generated sequences are highly likely to be extremely biased toward a specific subset of sequences
>
> Thank you, we agree that this posed an an important question that was not yet explicitly answered in the paper. We have conducted additional experiments to concretely demonstrate that our models are generalizing and that what is stated in the title is not the case.
>
> In short, we predicted structures for all 100K sequences in the DARK$_2$ training set allowing us to do two things. First we created a new training, validation, and test set split based on structure which we use to train and evaluate a new model (called DARK$_2$-_strict_), as is gold-standard in testing structure prediction methods and was recently suggested for design approaches in Ingraham et al. The test set sequences have no overlapping topologies with the validation and training set. Furthermore, any training/validation set sequences similar to test sequences have been discarded removed (Last paragraph of 3.3 and exhaustive details in Appendix B.2). The strict test set topologies are also included in the Appendix. As the results show in Table 1, DARK$_2$-_strict_ performs similarly to the other DARK models directly demonstrating that the model is assigning a high likelihood to sequences dissimilar to those its seen and to sequences with unseen structures. Secondly, we use this set 100K set to show that the DARK iterations generate sequences with unseen ordered topologies. We believe this provides a thorough and stringent demonstration of the generality of our approach.
>
> > ...Also, having 1000 clusters for 1000 sequences may suggest that this metric is not very reasonable to begin with. With that many sequences, one would expect at least a few clusters of size greater than 1 just by random chance (usually).
>
> Thank you, we agree that it is difficult to assign a precise quantitative amount to diversity from clustering results and we have clarified this in the text. The 30% sequence identity threshold is very common for clustering protein sequences (e.g. the widely used UniRef30 and the BFD). Lowering the clustering threshold below 30% sequence identity begins to increase the risk of incorrectly clustering sequences and we feel that the results become too noisy to provide value. Would you be able to clarify the last sentence above? The sequence space of $L=100$ proteins is massive, and even databases of clustered natural sequences like UniRef30 and the BFD have millions of singleton clusters. We are unclear as to why our synthetic sequences are likely to always have a few clusters greater than 1.
>
> > Furthermore, the authors claim that the 1000 samples drawn from DARK have no strong similarity to natural proteins in UniProt. This needs to be quantified.
>
> Thank you, we agree the explicit details should have been in the main paper, we originally left it in the appendix due to space constraints. We have moved this to the main paper and we have additionally included results from performing a search with the UF50 model samples.
>
> ---
>
> ### RE: The model is still conditioned on structure
>
> Respectfully, we disagree that our model is conditioned on structure. No DARK model is conditioned on structure, it is only provided sequences. Furthermore, learning a relationship between sequence and structure is neither conditioned out nor enforced in anyway. If the argument is that the model is limited to producing structures from the initial seed samples, we feel that is a question of generality and not the case, as is discussed above.

---

> > ### Author Response · Authors · 2021-11-27
> > **Response to Reviewer g4tX Cont.**
> >
> > ### RE: Outperforming natural sequences is concerning
> >
> > Respectfully, this is based on incorrect assertions for which we provide correcting information below:
> >
> > > ... because natural sequences (excluding those that are disordered) are typically highly thermostable and thus structured.
> >
> > Respectfully, this is incorrect. Natural sequences are not highly thermostable. Natural proteins are _marginally stable_ (Taverna & Goldstein, 2002) and it is one of their important attributes. The stability of natural proteins can be hugely affected by, in the most extreme cases, even a single mutation. As an aside, this is, among many other reasons, why the field of protein variant effect prediction receives significant interest. In contrast to natural proteins, a well known characteristic of _de novo_ designed proteins is that they are hyperstable and usually have 'ideal' structures (Huang et al., 2016). This is evidenced in that a majority of designs in the PDB to date are stable at 95C, a temperature at which most natural proteins denature (Baker, 2019). This was also shown for trDesign sequences, of those candidates tested by trDesign, all were stable above 70C and a portion were stable at 95C.
> >
> > > In fact, natural sequences in many ways should be the upper bound of how well the generating model performs (unless the claim is that these generated sequences are more structured than natural sequences)
> >
> > Respectfully, this is incorrect. Natural protein structures will not be the upper bound because they are marginally stable and have little clear structure information. This is well known in protein structure prediction and is clearly evident in the use of multiple sequence alignments as inputs by state of the art models. _De novo_ designed proteins are commonly known to have very easy to predict structures, especially given that they are predicted from a single sequence.
> >
> > We have added further detail in the Related Work as to why it is expected and known that unconditional models trained on natural sequences perform poorly. Our results for the UF50 model simply provide further evidence of this. We also highlight that Ingraham et al. also provides a clear demonstration of this using a rigorous structure-based test split.
> >
> > ---
> >
> > ### RE: There is no real validation of the generated sequences' structures
> >
> > > Using AlphaFold is unfortunately not a convincing validation of these sequences' structural likelihoods.
> >
> > Respectfully, in the context of protein design, asserting that structure prediction is not a convincing validation is incorrect. In protein design, predicting the structures of designed proteins is the be-all-and-end-all validation of a candidate (Huang et al., 2016). It directly tests the result of designing. Furthermore, to even consider a candidate for laboratory validation it needs to be confidently predicted to have a stable ordered structure. As such, it is the most important validation of sample quality.
> >
> > > ...it is still only an estimate with no theoretical guarantees. That is to say, it is not "real"
> >
> > Respectfully, there is no structure prediction method that we are aware of that provides a theoretical guarantee, so we are unclear on what is meant here.  AlphaFold provides us with an accurate, but not perfect, means to assess the most important aspect of testing design candidates: their predicted structure.
> >
> > > The model should give structured proteins a higher score than disordered proteins.
> >
> > We have added results showing DARK$_3$ assigning a higher likelihood to validated _de novo_ designed sequences over random sequences in Appendix B.4, but we stress that this does not tell us anything important for our applications. From a generative modelling perspective, given the pathologies generative sequence models can suffer from, it is imperative to directly assess sample quality. This is especially true as we use our generative models primarily for sampling. All accounted for, a likelihood estimate is far less important than assessing sample quality. Assigning a higher likelihood to validated sequences with a learned model is also not "real", if using a learned model is what is meant in this case?

---

> > > ### Author Response · Authors · 2021-11-27
> > > **Response to Reviewer g4tX Cont.**
> > >
> > > ### RE:  Missing some baselines
> > >
> > > Respectfully, we disagree that there are important baselines missing. We are aware of and had cited both works in our Related Work.
> > >
> > > **Regarding UniRep:** UniRep is a representation model, specifically for protein engineering. As discussed above and in the paper, models like UniRep fail to generalize beyond the structures represented by the sequences in its training set. For their purpose of Protein Engineering that is not of concern, but for _de novo_ design it precludes it as a method.
> > >
> > > >It should also be noted that UniRep is capable of generating sequences that are structured, which is shown in the paper.
> > >
> > > We have double checked the paper and the closest example we could find was Supplemental Figure 14, which shows a homology model generated by providing the first 15 amino acids of a sequence that is in its training set, although this is not clarified. It is also said to be >50% identical (potentially identical, but this is not clarified) to other family members which are also in its training set. We feel that this does not constitute an example of generating sequences with ordered structures in any general way as we do here. Furthermore, we struggled to correctly generate sequences with the publicly available model and similar difficulties have been reported elsewhere (Ma & Kummer, 2020).
> > >
> > > This all said, we feel that our UF50 model provides not only a direct comparison to a UniRep-like model, but a more fair one. We use the same training set, Uniref50, but it is more up to date and consequently double the size. We also use a transformer-based model, compared to the LSTM used in UniRep, which is also the same model used by DARK$_3$. It is also hyperparameter optimized on the synthetic validation set. Considering these factors, we feel this is as fair a comparison as we can reasonably make.
> > >
> > > **Regarding Anand & Huang (2018):** Their work explores generating protein backbone coordinates with GANs. More specifically, they suggest a fast method for loop modeling. Although they extend their approach to generating sequences with Rosetta from their generated backbones, it is not their main contribution, and the structures of those sequences are extremely disordered, as seen in their Figure 3, B-F. We note Figure 3 A includes real proteins with significant disordered segments too. Considering by comparison the clear order of the structures in our Figure 5 and that their work focuses on a different problem, we feel it is it is not comparable for benchmarking purposes.
> > >
> > > ---
> > >
> > > ### RE: Some choices in the algorithm are unclear or of questionable soundness
> > >
> > > > The choice of the loss function is not quite clear. For example, why use the background model p(y)? Why not just maximize p(y|x)? Additionally, is the sequence prior needed? What happens if it is removed? Is there a weight \gamma for these two halves of the loss function?
> > >
> > > Regarding the background model, foremost of reasons for our choice is the demonstrated prior art of trDesign. Secondly, a similar approach using a background model was used to great effect in the first AlphaFold paper (Senior et al. 2020) in the context of a distance potential for protein structure prediction. Furthermore, the way we use the information gain/relative entropy/KLD as we do is formally grounded. Regarding directly maximizing $p_{\beta}(y_{d}|x_{d})$, we are using the information gain as a way of indirectly optimizing $p_{\beta}(y_{d}|x_{d})$ because it is unclear how it could be directly optimized in this setting; we are not providing specific structures for which we want to optimize $p(y_{d}|x_{d})$. That said, if you have any suggestions or comments as to how to directly optimize $p_{\beta}(y_{d}|x_{d})$ we are greatly interested.
> > >
> > > Regarding the sequence prior, thank you for the suggestion. To paraphrase another response, we previously explored different weighting schemes between the two and did not find it to provide a clear benefit. We also found that the sequence prior provides a way to numerically check if a model collapses, although we did not see any collapse in our trained models. Thus, for clarity we do not explore it in the text, but it is still of interest for further work.
> > >
> > >
> > > > It is also not clear why examples are concatenated (step 6 of the algorithm). Do old examples need to be present even though larger and larger sets of sequences are being generated?
> > >
> > > We feel removing examples runs counter to the underlying data availability problem that DARK iterations tackle. The aim is to continue to grow a bigger and bigger dataset. There are no "old examples" as they are all valuable training data. Removing sequences from the growing set is the opposite of our aim so we are unclear on what is intended by the question, could you clarify what is meant?

---

> > > > ### Author Response · Authors · 2021-11-27
> > > > **Response to Reviewer g4tX Cont.**
> > > >
> > > > > The use of a different model architecture for each iteration of DARK is concerning. This suggests that the developed method may be extremely overfit to this specific task/dataset. There doesn't seem to be a justification for the different architectures.
> > > >
> > > > We assure you this is not any sign of being extremely overfit. Quite to the contrary, the various results we present demonstrate that our models are in fact generalizing. The use of a different model is quite benign. It was entirely decided by the fact that the dataset increases by a order of magnitude between iteration one and two (15K to 100K examples); a larger model for a larger dataset. We have added explanation to the relevant section.
> > > >
> > > > > Also why are Cys residues removed? Given that disulfide bridges are a critical part in maintaining tertiary structure in many proteins, a model that does not consider their existence raises some red flags.
> > > >
> > > > We assure you this is in no way a 'red flag'. Considering your comment and another reviewers, we feel that we did not provide sufficient background from the field of _de novo_ protein design as to why not including cysteines is common and is not of note. We have added further discussion to both the text and in the Appendix (A.7).
> > > >
> > > >
> > > > ---
> > > >
> > > > References:
> > > > Taverna DM, Goldstein RA. "Why are proteins marginally stable?" _Proteins_, (2002).
> > > >
> > > > Huang, PS., Boyken, S. & Baker, D. "The coming of age of de novo protein design." _Nature_, (2016).
> > > >
> > > > Baker, D. "What has de novo protein design taught us about protein folding and biophysics?" _Protein Science_, (2019).
> > > >
> > > > Senior, A. W., Evans, R., Jumper, J., Kirkpatrick, J., Sifre, L., Green, T., ... & Hassabis, D. . "Improved protein structure prediction using potentials from deep learning." _Nature_, (2020).
> > > >
> > > > Alley, E. C., Khimulya, G., Biswas, S., AlQuraishi, M., & Church, G. M., "Unified rational protein engineering with sequence-based deep representation learning." _Nature methods_, (2019).
> > > >
> > > > Ma, E. J., and Kummer A., "Reimplementing Unirep in JAX." _bioRxiv_ (2020).
> > > >
> > > > Anand, N., & Huang, P. S. "Generative Modeling for Protein Structures." _NeurIPS_, (2018).

---

> > > > > ### Comment · Reviewer_g4tX · 2021-11-30
> > > > > **Reply**
> > > > >
> > > > > ### Re: Generated sequences are highly likely to be extremely biased toward a specific subset of sequences
> > > > > It would be important to quantify that the method can generate topologies outside of the 100k seed sequences/training set. If this can be quantified and shown, then I would be more satisfied with this point.
> > > > >
> > > > > ### Re: Outperforming natural sequences is concerning
> > > > > Point taken. I would be nice to see how natural sequences lie in the distribution of generated sequence with respect to thermostability, if possible.
> > > > >
> > > > > ### Re: The model is still conditioned on structure
> > > > > Importantly, DARK is reliant almost entirely on DMPfold2, which is a _supervised_ model that learns based on structural labels. So even though DARK doesn't take in protein structures as an explicit input, information about structural labels is still coming in through DMPfold2. A model that is truly unconditioned on structure would not take in structural labels either directly or indirectly through a supervised model. This is an important distinction to make, as calling this method unsupervised/unconditioned on structure is misleading.
> > > > >
> > > > > ### Re: There is no real validation of the generated sequences' structures
> > > > > AlphaFold, although impressive, is _not_ truth. In fact, we have already seen it fail in many ways, especially on non-natural sequences (namely sequences of folds of topologies that are rare or underrepresented in the AlphaFold training set). Given that the goal of this submission is to generate novel protein designs (of perhaps some novel folds), it becomes especially unrealistic to rely on AlphaFold as a gold-standard oracle.
> > > > >
> > > > > For this particular work, the main contribution is essentially to use a neural network (DMPfold2) to generate structures which are then evaluated by another neural network (AlphaFold). Since there are no experimental validations that are provided, it is not convincing that DARK can generate novel folds that are in any way realistic, other than to trust AlphaFold. This is unfortunately not very convincing in my opinion.

---

> > > > > > ### Author Response · Authors · 2021-12-03
> > > > > > **Further Response to Reviewer g4tX**
> > > > > >
> > > > > > ### RE: Re: Generated sequences are highly...
> > > > > >
> > > > > > In 4.1, the final paragraph titled _"DARK models generalize to sequences with unseen structures"_, this is quantified in the revised manuscript, along with the results of the DARK$_2$-_STRICT_ model, with the increase in topologies between the 15K seed set and the  85K set from the first iteration of DARK. We are unclear if this is also asking to repeat the same validation between this combined set and the next iteration of DARK. If so, we are not clear as to how it significantly adds to the current analysis or any conclusions drawn. We feel the DARK$_2$-_STRICT_ results and the topology examination both separately provide a strong demonstration of generality, and together even more so.
> > > > > >
> > > > > > ### RE: Re: The model is still conditioned on structure
> > > > > >
> > > > > > Respectfully, we continue to disagree that our models are conditional. We feel that the argument posed incorrectly conflates structure information entangled in the sequence with separately provided structure labels. This is a key distinction; providing structure labels conditions out structure information from the problem. In contrast, the onus is on an unconditional model like DARK to learn a sequence distribution that disentangles structure information in some latent way, as is the case here. It is desirable but it is up to the model whether or not it does this. DMPfold2 as a supervised model is an approximation of the ground truth mapping of sequence to structure that nature ‘knows’. The sequences in our work are optimized with DMPfold2 and natural sequences are optimized by nature. By conflating labels and entangled information, training on natural sequences should also be incorrectly considered conditional, especially as nature has access to the exact mapping and labels.
> > > > > >
> > > > > > We feel it is accurate to state that DARK models are trained unconditionally on sequences, and those sequences are optimized by the IG-score using DMPfold2 without labels. We also feel that empirically we have shown that DARK models are learning a sequence distribution that is general with regards to 'unseen' structure, providing an empirical indication that this approach is unconditional in that regard.
> > > > > >
> > > > > > > ...as calling this method unsupervised/unconditioned on structure...
> > > > > >
> > > > > > We are confused as to the conflation of unsupervised and unconditional, as used here. By extension, we are unclear as to how a DARK model or the IG-score optimization can be considered supervised.

---

> > > > > > > ### Author Response · Authors · 2021-12-03
> > > > > > > **Further Response to Reviewer g4tX Cont.**
> > > > > > >
> > > > > > > ### RE: Re: There is no real…
> > > > > > >
> > > > > > > Regarding AlphaFold, whether or not it is a gold-standard could be a matter of debate, however, nobody has produced a more accurate structure prediction method to date. This includes the Baker Lab (the lab behind Rosetta), whose structure prediction methods are those typically used to validate computational _de novo_ designs (for example, see our use of Rosetta AbInitio Relax or trRosetta in trDesign). There has only been a small number of _de novo_ designs released in the PDB outside of the AlphaFold training set, as can be seen in our manuscript, so outside of the examples in CASP14 it is difficult to provide a strong statistical determination of its success in particularly folding _de novo_ designed sequences. That said, it has been seen that AlphaFold provides accurate and confident predictions for that small number of examples, as would be expected.
> > > > > > >
> > > > > > > > For this particular work, the main contribution is essentially to use a neural network (DMPfold2) to generate structures which are then evaluated by another neural network (AlphaFold).
> > > > > > >
> > > > > > > Respectfully, this is incorrect and a mischaracterization of our work. This disregards the DARK models and all primary contributions as explicitly stated and discussed in the manuscript, including the DARK framework and AlphaFold Refinement method. Furthermore, DMPfold2 is not used to generate structures. As a part of our work, it is used in an optimization procedure to generate or refine sequences. Also, DARK$_3$ is used as the final design tool that generates sequences for different applications.  This statement also disregards the interesting connections, among others, between our approach and similar approaches in model based optimization literature and the potential machine learning methods that could be developed building on our approach.
> > > > > > >
> > > > > > > >Since there are no experimental validations that are provided, it is not convincing that DARK can generate novel folds that are in any way realistic, other than to trust AlphaFold.
> > > > > > >
> > > > > > > Our work uses the best available structure prediction method to select candidate designed sequences, the standard and well proven approach in computational _de novo_ protein design. We feel that to disregard it on that basis runs contrary to what has been demonstrated in the field. As we say in the manuscript, the goal of a performant computational protein design algorithm is to efficiently provide as many high confidence candidates as possible to justify the high cost and high attrition rates of laboratory structure determination. Structure prediction is the de facto approach to computationally validate sequence designs in _de novo_ protein design. Also, we have not stated that our primary goal is to produce sequences with novel folds, however, we have demonstrated that DARK iterations both generate novel (being unseen to the models) structures and that DARK models place a high likelihood on sequences with structures unseen. These indicate that this is possible.
> > > > > > >
> > > > > > > Even ignoring these points, there is direct evidence that methods like ours and trDesign are capable of producing sequences with highly ordered and realistic structures. Namely, Anishchenko _et al._ demonstrate with trDesign, as recently published in Nature, several crystal and NMR structures of just such designs.
> > > > > > >
> > > > > > > ---
> > > > > > > References:
> > > > > > >
> > > > > > > Anishchenko, I., Pellock, S.J., Chidyausiku, T.M. et al. “De novo protein design by deep network hallucination.” _Nature_ (2021).

---

### Official Review · Reviewer_jJy9 · 2021-11-01

**Correctness:** 3
**Technical Novelty And Significance:** 2
**Empirical Novelty And Significance:** 2
**Recommendation:** 5
**Confidence:** 4

**Main Review:**

Overall, the paper is well written and easy to follow. The main ideas are clearly explained, the contributions and the empirical evaluation protocols and results are well presented.

On the strenghts side, the presented method make use of strong state-of-the-art components that are assembled to provide an end-to-end approach for de novo protein sequences and structures design. The comparison with similar approaches is well motivated and seems to be fairly adressed.

On the weakness, I would be happy to have feedback of the authors on the following questions:
1) You state in section 3.1 that you do not consider Cysteine for experimental reasons? Why? Can you elaborate on the implied limitations of this choice?
2) You present your work as a faster alternative to the trDesign approach. Why is there no computational times reported? How long does it take to your approach to match the perplexity or IG-Score of the seed sample generated by trDesign?
3) Why do you mesure the IG-score on 1000 samples only?
4) Section 4.2: this sentence is really difficult to understand without having to look to the appendices "For further context, we calculated the mean pLDDT scores of recently released de novo designed sequences out of AlphaFold’s training set and found it to be high at 88.1 (σ = 4.41). This suggests that Good+ pLDDT is a good measure of a model’s ability to produce samples with a quality sufficient for design tasks."

Typos:
- Page 3: "develped"
- Page 8: "with with"

**Summary Of The Paper:**

The paper present a method to sample jointly de novo protein sequences and structures without relying on any structure conditioning. The main contribution is a new iterative procedure to generate a dataset of increasing size in an unsupervised fashion. This dataset is finally used to learn a generative model that captures the distribution over the "unnatural" sequence space, structures are predicted by using AlphaFold on the generated sequences. The authors provide empirical evidence of the quality (diversity and structural stability) of the de novo designs.

**Summary Of The Review:**

The paper is clearly of interest. The approach is, up to my knowledge, a novel and wise way of assembling building blocks from various recent articles. However, in the current state of the article, there are still a few points that need to be clarified in order to be fully equiped to assess the potential of the approach.

---

> ### Author Response · Authors · 2021-11-27
> **Response to Reviewer jJy9**
>
> Thank you for your feedback and comments. Please see our answers to each question below:
>
> > 1.You state in section 3.1 that you do not consider Cysteine for experimental reasons? Why? Can you elaborate on the implied limitations of this choice?
>
> Thank you, we agree this should have been explained in more detail and we have added further explanation to the text in 3.1 that additionally links to further detail in Appendix A.6. In short, not including cysteine is common in _de novo_ protein design studies and does not pose any significant limitations to our work or the conclusions drawn from it. It also provides a benefit. Its inclusion makes expression of any designed sequence in bacteria (the most common way to produce designs in the lab) more challenging.
>
> Cysteine is special among amino acids, as it is mainly occurs in proteins to form disulfide bonds, particularly secreted proteins and outwards facing membrane proteins. This happens between pairs of cysteines, and forms a strong, usually stabilizing, bond between them. This bond cannot form in the cytosol of the cell, so disulfide containing proteins are expressed in an otherwise different way. This is the root cause of why it complicates validation of designed sequences. In _de novo_ design literature the inclusion of cysteine is typically discussed as the inclusion of disulfide bonds. _De novo_ designed proteins are known for being hyperstable without disulfides, and it is often a question of if the inclusion of disulfide bonds is justified. In other words, its common that the default stance is to not include disulfide bonds or to add them to preexisting candidates. For example, the Rosetta software package has separate protocols explicitly for adding disulfides to designs. Similarly to trDesign, we don't feel that including them here provides a clear benefit compared to the complications it brings.
>
> ---
> > 2.You present your work as a faster alternative to the trDesign approach. Why is there no computational times reported? How long does it take to your approach to match the perplexity or IG-Score of the seed sample generated by trDesign?
>
> Thank you, we agree this should have been a part of the text. We have added significant timing information throughout and we believe it now forms a part of our argument. DARK$_3$ without refinement can produce samples with similar IG-scores to our version of trDesign (DARK$_0$ ) in 0.05 seconds. In contrast, DARK$_0$ takes over 11 minutes (Table 1).
>
> ---
> > 3.Why do you mesure the IG-score on 1000 samples only?
>
> Our decision to use 1000 samples to measure the IG-score was motivated by the time it takes to calculate the pLDDT scores. For consistent comparison, we calculate both metrics on the same set of 1000 sequences. A larger number of sequences would not noticeably effect the IG-score calculation time but it would have made calculating the pLDDT more difficult with the resources available; 1000 sequences predicted on a single graphics card is approx. 5 hours. We feel that the decision is justifiable because, of the values investigated, using larger samples of sequences did not significantly change results or effect the conclusions drawn.
>
> ---
> > 4.Section 4.2: this sentence is really difficult to understand without having to look to the appendices "For further context, we calculated the mean pLDDT scores of recently released de novo designed sequences out of AlphaFold’s training set and found it to be high at 88.1 (σ = 4.41). This suggests that Good+ pLDDT is a good measure of a model’s ability to produce samples with a quality sufficient for design tasks."
>
> Thank you, we agree this is unclear. Where we have discussed these results we have tried to make the text more clear.

---

### Official Review · Reviewer_mSTZ · 2021-11-02

**Correctness:** 3
**Technical Novelty And Significance:** 2
**Empirical Novelty And Significance:** 2
**Recommendation:** 3
**Confidence:** 4

**Main Review:**

The paper provides an effective technique for generating sequences that are predicted to fold into a stable structure without any conditioning. The authors focus on de novo protein design which is left without a precise definition by the authors but appears to refer to designing sequences that fold into structures and are not "similar" to natural sequences. No further details about what similarity means or how it is measured are given. I personally find the problem a bit underspecified and it is not clear what a practical application of a method that generates sequences that are predicted to fold but are otherwise completely random is. Can the authors please explain further what the concrete goal is and what the application is?

The primary issue is that there is a lack of available training data. The authors therefore apply a technique used in trDesign to generate synthetic training data that they hope fit their criterion of folding to a stable structure. The final method iteratively samples from the generative model, optimizes the sequences such that they are predicted to fold using the trDesign approach, and then retrains the model on the generated sequences.

While the method works, it appears to mostly be bootstrapping on the method proposed by trDesign. To put it another way trDesign is used to generate lots of sequences and then the authors train a language model on those sequences and demonstrate the resulting sequences are able predicted fold ~30% of the time by AlphaFold and are reasonably diverse. Most of the technical contribution seems to come from trDesign. There is nothing unique to the approach that encourages diversity or even consistency with respect to the distribution that they are trying to learn. It appears mostly to iteratively retrain itself on inputs generated by trDesign.

While DARK improves over two baseline models, gcWGAN and gramVAE, according to their new pLDDT metric, these methods use different oracle functions for structure prediction. It is possible that the entire difference can be explained by using DMPfold2 rather than the structure prediction method in gcWGAN. As the paper currently stands, I do not find these comparisons compelling.

In order to improve the paper the authors should
1) Provide a more concrete definition of de novo design and more clearly state what the goals of this paper are and why trDesign does not achieve those goals. Why do we need an unconditional model to generate sequences? What application does this solve that trDesign could not already solve?

2) Better baselines could be performed by implementing other methods to use the same structure prediction oracle.



**Summary Of The Paper:**

This paper presents an unconditional generative model that can produce sequences predicted to fold into stable structures.

**Summary Of The Review:**

Overall, the authors present an interesting method that clearly achieves their goal of generating sequences predicted by AlphaFold to fold. I am left feeling that this problem is not clearly motivated and that the solution is not significantly novel from an ML perspective.

---

> ### Author Response · Authors · 2021-11-27
> **Response to Reviewer mSTZ**
>
> Thank you for your comments and feedback. We answer the first point foremost to hopefully aid in answering the points that follow:
>
> > 1. Provide a more concrete definition of de novo design and more clearly state what the goals of this paper are and why trDesign does not achieve those goals. Why do we need an unconditional model to generate sequences? What application does this solve that trDesign could not already solve?
>
> Thank you, we feel we have significantly revised and refocused the paper, particularly, we hope, to answer these questions more clearly. We agree that the submitted version of the paper did not address these clearly enough and lacked a cogent flow in that regard. Revisions are particularly evident in the introduction, first two parts of the Related Work, and 3.2. We have also included significant additional results not just regarding the models but also their applications.
>
> ---
>
> > No further details about what similarity means or how it is measured are given
>
> We agree that explicit values should have been in the main text. We have moved the sample similarity results to the main text. Sequence similarity and searching are mature problems in the field of Bioinformatics. Similarity is usually measured by a simple sequence identity metric, or by the probabilistic E-value produced by a search tool like BLAST, MMseqs2, or HHblits.
>
> > Can the authors please explain further what the concrete goal is and what the application is?
>
> To paraphrase the revised text, our aim is to develop a tool, based on a deep generative model, that can rapidly generate sequences with stable and ordered structures. We can use it to rapidly generate candidates for common and high-interest topologies. We can also use it to help develop further novel design approaches. We have added additional results and methods for such a novel design approach.
>
> > The authors therefore apply a technique used in trDesign to generate synthetic training data that they hope fit their criterion of folding to a stable structure... Most of the technical contribution seems to come from trDesign.
>
> Respectfully we disagree, however we agree that this was not explained properly on our part. We have clarified and added to the methods and results, comparing trDesign and our approach. To add further here, we are able to build a model that generates (without any refinement) similar sequences to trDesign but orders of magnitude faster. Our iterative approach to building this model is our own. Furthermore, the greedy refinement we do is small compared to the simulated annealing done in trDesign (3000 steps vs 40,000 steps). It improves upon the samples from our generative model. Would you be able to elaborate on your comments regarding "encourages diversity or even consistency"? We have focused on showing the generality of our approach. We have added results to show that our model learns to generalize from a structure perspective and that DARK iterations introduce sequences with topologies unseen in the initial seed samples. It is unfeasible with resources available to us to sample the huge number of sequences needed to accurately characterize the distribution of a trDesign-based method, and we are unclear as to the benefits of attempting to exactly mimic it.

---

> > ### Author Response · Authors · 2021-11-27
> > **Response to Reviewer mSTZ Cont.**
> >
> > > ...these methods use different oracle functions for structure prediction... It is possible that the entire difference can be explained by using DMPfold2 rather than the structure prediction method in gcWGAN.
> >
> > Would it be possible for you too elaborate further on this? For the revised version of the paper we have moved the results for the gcWGAN and the gramVAE to the Appendix (Table 8; pg 26.). We originally provided both as an attempt to demonstrate how our model succeeds even over methods, even with conditioning information. However, upon reflection of your comments and the other reviewers we feel that the two methods are advantaged over ours too a point that makes the comparison unclear enough to detract from the clarity of the analysis more than add to it.
> >
> > As previously discussed, both models have access to structure-based conditioning information. Additionally, both are trained on natural sequences that have known structures and are all in AlphaFold's training set. If either of them recapitulates a sequence from their training sets, or one similar, AlphaFold will be heavily biased towards predicting the original structure and doing so with a strong confidence, boosting the pLDDT. Several methodological details in both models are also unclear.
> >
> > Further, we feel that the wcGAN is even more advantaged because it has a fold predictor directly in its training loop. Its generator can be actively corrected as it learns. Our model never has access to DMPfold2 during training, its a standard autoregressive training scheme.
> > We argue that one of the novelties of our approach is the data itself, or rather, training a model on this kind of synthetic data - the sequences are optimized to have a clear signal for structure, but no specific structure. Our model doesn't need to learn a sequence distribution that incorporates that information, but our results show that it does. Natural sequences, on the other hand, do not have this clear structure signal. We feel that this lack of structure signal from training on natural sequences to be a critical point and we have added further details of this in the Related Work.
> >
> > ---
> >
> > > 2. Better baselines could be performed by implementing other methods to use the same structure prediction oracle.
> >
> > As mentioned above, we must admit to some confusion with regards to what is intended. That said, your question prompted us to investigate a possible method that might provide insight to what you ask, which we now include as DARK$_1$-Adversarial (Table 1; pg. 6). This is the same as the DARK$_1$ model however we include DMPfold2 in the training loop with the IG-score. DMPfold2's parameters are frozen but, with a straight-through estimator, we pass it a one-hot sequence sample from the transformer decoder, and then use them to calculate the negative IG-score which is incorporated into the loss. It is briefly discussed in the main text, but we go into the finest detail in the Appendix if it is of interest (A.3; pg. 16). The IG-score, and other metrics, is very similar to the original DARK$_1$ (Table 1 & 8) which we suspect is simply due to either DMPfold2 simply acting as a source of noise, or that it was simply reinforcing the same loss signal that the training sequence was already providing.

---

> > > ### Comment · Reviewer_mSTZ · 2021-11-27
> > > **Response to confusion regarding review**
> > >
> > > To elaborate, your method uses DMPfold2 as a structure prediction oracle. The oracle plays a significant role here because your validation relies on AlphaFold deciding how confident it is that a sequence folds to a particular structure. If you use a bad oracle, then even if your algorithm performs 100% as desired, it is being trained to generate bad sequences due to the oracle. Methods like gcWGAN use their own sequence->structure oracle and my comment was pointing out that it is difficult to do an apples-to-apples comparison to sequence design methods that use different oracles.
> > >
> > > Ideally you would compare all of these methods when the same oracle, e.g. DMPfold2 is used, in order to separate the algorithmic contributions from the improvement you can get my simply reimplementing a method like gcWGAN with a different oracle.
> > >
> > > I am confused by your comment "Our model never has access to DMPfold2 during training, its a standard autoregressive training scheme."
> > > Line 5. of Algorithm 1. says Rapid optimization of x_s IG-score using the oracle. Line 6 adds this optimized sequence to the list of sequences that the model is then trained on. Maybe we are disagreeing just about semantics, but perhaps I am really misunderstanding the procedure because it seems that DMPfold2 is used during training (specifically on Line 5).

---

> > > > ### Author Response · Authors · 2021-11-29
> > > > **Response to response**
> > > >
> > > > Thank you for clarifying. Touching on your last point first, we consider Line 3 the training of a given model because that's the only point where it's updating its parameters and learning. During that step, the model, for a given N, is trained normally on the current training set. The samples it generates that are refined and added to the list, lines 4-6 as you say, are only relevant to the training of the next iteration's model. So for example, DARK$_2$ trains on its 100K sequence set and then it's frozen. The 400K samples it generates, which are refined and added to the set, are only relevant to the DARK$_3$ training. It might also be worth re-clarifying that the model parameters at each iteration are randomly initialized.
> > > >
> > > > The models have access to the refined sequences, which are the product of DMPfold2 optimization, but they're not 'aware' of this provenance as it's just a static training set. In contrast, we'd say a model that does have access to its oracle during training is DARK$_1$-_Adversarial_. I suspect you're right in saying that we are on the same page but disagreeing about semantics, but on the chance we aren't hopefully this helps clarify?
> > > >
> > > > Regarding the topic of the oracle comparisons, yes we agree the oracle dictates the upper bound on how well you can expect your model to perform WRT AlphaFold's validation of samples. The difficulty we see with simply re-implementing a conditional model like gcWGAN with a different oracle like DMPfold2 is that its still a different problem setting (structure conditional with natural sequences) and it's not an apples-to-apples comparison. To be comparable, the approach would need to train on synthetic sequences and remove its conditional structure information. At a minimum, it removes the bias of having the entire training set covered by AlphaFold's training set, and removes the aid from getting conditional structural information. Because of the way they use their oracle to reinforce that conditional information, without that information a different approach needs to be used to incorporate the oracle's prediction in the loss, like maximizing IG-score. DARK$_1$-_Adversarial_ is an example of implementing one of the many interesting ways of doing that. At this point though we feel the comparison method has become a completely new approach itself. Unlike gramVAE/gcWGAN, the oracle is the only thing DARK gets to provide information to the models and in that setting the comparison of contributions is unlikely to be interesting in so far as better structure predictors likely produce better sequences.
> > > >
> > > > If the question is a matter of exploring the performance of different $p(x)$ or $p(z)$ generator architectures in a DARK-like setting with the same DMPfold2 oracle then we agree its certainly interesting to explore beyond the RNN comparison we include. But, we feel that is the topic of following work. Here we are focusing on trying to show that our approach works, and it has practical applications, and we feel this already introduces significant complexity to the results we present. Looking forward, we feel that our work in part shows that learning with synthetic protein sequences for _de novo_ design is a viable setting for method development and that there are many exciting potential ways to do it.  Hopefully this helps make things clearer? And thank you again for clarifying.

---

### Official Review · Reviewer_sSh9 · 2021-11-03

**Correctness:** 3
**Technical Novelty And Significance:** 3
**Empirical Novelty And Significance:** 2
**Recommendation:** 5
**Confidence:** 3

**Main Review:**

The larger methodological message of the paper seems to be that, if your goal is to generate samples of sequences in proportion to their score, rather than "start from N random sequences and anneal them to your objective", you can start by annealing a small number of random sequences, then that first batch to guide the generation of new (non-random) initial sequences to anneal. Eventually you can work your way up to N total annealed sequences, but most of them started with better initializations, so the hope is that the overall scores are better.

That's a reasonable thing to propose and to try in the context of protein sequence design. But I have moderate concerns about how this was depicted, because the paper doesn't really try to connect or contrast this scheme with larger ideas from sampling (e.g., "estimation of distribution" methods) or from machine learning (e.g., self-training). Even if those connections were made clear, ultimately I also find the choice of baselines confusing, because they are all so different from they don't get to the heart of what's different about this method, i.e., too many things change between the compared methods.



### Comments

I think the high-level methodology here can be viewed conceptually a kind of self-training within an EDA ("estimation of distribution") framework [see Figure 3.2 in Larranaga and Lozano 2002, Estimation of Distribution Algorithms: A New Tool for Evolutionary Computation]. The paper would benefit if it were framed that way, and if it discussed success / failure modes of self-training in scenarios like sequence generation [He et al. 2020, Revisiting Self-Training for Neural Sequence Generation], classification, etc.

Viewing DARK as an EDA/self-training of trDesign, I think also "stDesign" for "self-training design" would have been a more fitting name, attributionally, than something completely new like DARK. But that's OK.

The paper acknowledges multiple times that it builds upon Anischenko, and that's great. But the way the paper is written it often still fails to make clear what is new and what is borrowed. That is especially true of section 3.4, which is describing Anischenko's objective and proxy objective, but in different notation, and without making this clear.

One of the listed contributions is "By pairing DARK with AlphaFold, we show a fully neural way to unconditionally sample both protein sequences and structures jointly." Just some feedback on that: I think it's a bit of a stretch to call this a contribution. First, spart from self-training, the overall joint sampling scheme is essentially taken from [Anishchenko et al 2020]. [Anishchenko et al 2020] also mention that annealing could in principle be replaced with gradient ascent by backpropagating to the sequence, akin to which others like [Killoran et al 2017] attempted in simpler settings, which seems more "neural" than what the submission is doing.

The first listed contribution says that this is the first paper to "show that unconditional generative models... can capture the important long range dependencies that relate protein sequences with their structures." As far as I cans ee from Figure 2, it recapitulates alpha helices and beta sheets, of varying lengths; that's good, and you can probably argue that these are indeed long range (not purely pairwise, for example). But, you don't need deep learning to sample from the space of protein sequences that are broken up into stretches of beta sheets and/or alpha helices. Could the authors please comment on whether other structures were observed in sampling.

In most protein design situations, there is obviously a design criterion (affinity to a target, fluorescence, etc.), and that design criteron may call for conditioning on a family or families of proteins, or on desirable features of the protein sequence. Randomly sampling proteins that have "order" casts a relatively wide net, so I think it'd be helpful for the authors to talk about how they envision an unconditional sampler being used with respect to realistic design criteria. Are the authors envisioning to simply reject the ones that fail the criterion, from a huge pool of randomly sampled (but ordered) proteins? [Anishchenko et al 2020] go to the trouble of experimentally validating the secondary structures of predicted sequences, but they also do not express a clear vision in this regard.

By analogy to \beta-VAEs [Higgins et al 2017, ICLR], there might be a missed opportunity to tune the sample quality by introducing a simple hyperparameter that controls the weighting between the two KL terms.

The paper repeatedly talks of producing "high-quality structures" when it effectively means something more like "highly ordered structures" or "structures with high AlphaFold confidence." Whenever there's a specific notion of quality being pursued, the wording should reflect that, and it's a more fair way to characterize the situation when comparing to other generative methods that had different goals and therefore different notions of 'quality'.

Do the samples depicted in Figure 2 represent, proportionally, the kind of samples generated?

I am surprised that the paper does not include a "DARK_0" baseline, which would essentially be just the first round of [Anischenko et al. 2020] using the alternate representations.

Since DARK includes multiple rounds, and each round includes both training and hill-climbing, it seems fair to ask whether applying the same computational budget to "annealing from purely random seeds" baseline would produce comparable results and diversity. On some level, that is the most natural baseline (and other variations of it that would exhibit different diversity/score tradeoffs, e.g., different rejection thresholds, or pure hill-climbing). If the authors could please comment on this.





### Minor comments

page2: "to unconditional sample" -> "to unconditionally sample"

page2: please remove "rigorously"; the results in this paper are suggestive in nature

page3: "that does considers" -> "that does consider"

page4: "by Bayes" -> "by chain rule"

page4: Saying that one of these quantities

page4: Symbol \mathcal{X} seems described as a "sequence space" (which V^L is) and is then treated as a set of training points. I think the sentence introducing \mathcal{X} and \mathcal{Y} can be tidied up.

page6: "p_(y_d" -> "p(y_d"

page6: "distills" now has a specific technical meaning in deep learning so just be careful to avoid confusion.

Figure 2 is not referenced anywhere.

**Summary Of The Paper:**

I think it helps to start by summarizing [Anischenko et al. 2020, Protein design by deep network hallucination], and then explain the differences in the submission.

[Anischenko et al. 2020] aim to generate AA sequences that are diverse but that still reflect the properties of training structures.
* Their goal is to generate protein sequences whose structures tend to be unlike those of random sequences. Since random sequences have structures that tend to be unordered, they incidentally generate sequences having ordered structures.
* They use the deep resnet from rtRosetta (similar to AlphaFold's) to predict interresidue geometry maps from sequence, and that is how structure is represented and compared.
* Their design goal is to design samples from p(structure | sequence) p(sequence).
* To represent p(structure | sequence) they use a proxy objective: the KL divergence between the predicted structures and the structures of random sequences; in other words, good sequences are those whose distance maps are very different from the distance maps of random sequences.
* To represent p(sequence) they just use -KL(AA frequences of predicted sequences || AA frequences of PDB sequences).
* Sequences are designed by starting from random and then annealing according to the proxy objective.
* Designed sequences are then clustered by template modeling score (TM-score) and selected (from each cluster) based on  a measure of prediction 'consistency'.


Now, the submitted manuscript (DARK) is the same objective and proxy objective as [Anischenko et al. 2020], but with following methodological changes:
1. For the sequence->structure oracle, instead of using rtRosetta's interresidue geometry maps as the structure representation, here distograms [Senior et al. 2020] are used, and are output from a modified version of DMPfold2.
2. Rather than relying solely on one round of "random initialization -> annealing", DARK uses an iterative self-training scheme. First they do one pass of [Anischenko et al. 2020] to get 1st round of annealed sequences. The annealed sequences from 1st round are used to train an autoregressive generative model (Transformer-based). That model is sampled from to generate additional "random initializations" (which are now guided, not purely random). These guided samples are then optimized (just hill-climbing now instead of annealing, for speed), and added to the training set for subsequent rounds, etc.

There are other differences from [Anischenko et al. 2020], like using pLDDT score (orderedness) as a measure of "success", but the above two seem to be the main differences. The first of these two is worth trying but is not a conceptual contribution. The main idea of the paper is the second: self-training to generate diversified 'intermediate' initializations for further refinement. Basically it's using the initial refined sequences as a self-prior for exploring new sequences to refine.

**Summary Of The Review:**

The core methodological idea is worth exploring, and ultimately publishing, in the context of sampling protein sequences. But, the novel part of the methodology is conceptually similar to other population-based sampling strategies like EDA (estimation of distribution) and to self-training, and those connections are not mentioned or contrasted. Importantly, the baselines are different in ways that aren't relevant to the core novelty, making it hard to draw strong conclusions about the core idea itself (versus the influence of all the other maybe-helpful-but-not-methodologically-interesting changes in the system proposed). I'm open to changing my rating, but this is my take.

---

> ### Author Response · Authors · 2021-11-27
> **Response to Reviewer sSh9**
>
> Thank you for your feedback and comments. Please see our responses individually below:
>
> > I think the high-level methodology here can be viewed conceptually a kind of self-training within an EDA... But, the novel part of the methodology is conceptually similar to other population-based sampling strategies like EDA (estimation of distribution) and to self-training, and those connections are not mentioned or contrasted.
>
> Thank you, we are very concious of this connection and we agree that there is a strong conceptual similarity. When we began work we initially considered focusing on a paper with that perspective. We did include the following line in the first submission as a recognition of this connection, "We note that the iterative aspect of DARK shares similarities to methods like Estimation of Distribution Algorithms (EDAs), a common approach in model based optimization (MBO) (Bengoetxea et al., 2001; Brookes et al., 2020)."(pg. 5-6) but we fully agree that is does not fully explore that connection. We have added to this to saying that the connection should be explored in future work.
>
> We agree that exploring that connection properly is important. To do it justice, we see that as a follow up work to what is presented here. We feel that, particularly with our focus on _de novo_ design, it is imperative to first show a method that works, in a relatively simple form, and has applications in a practical sense. For this reason, we also approached choosing hyperparameters, particularly for DARK, as simply as possible, especially given how many different facets there are too this work.
>
> > But the way the paper is written it often still fails to make clear what is new and what is borrowed.
>
> Thank you, we agree. We have made significant changes to 3.2 and elsewhere. We hope this much better explains what is new and borrowed, as well as the explicit reasons for how our approach succeeds where trDesign does not.
>
> > One of the listed contributions is "By pairing DARK with AlphaFold, ...  Just some feedback on that: I think it's a bit of a stretch to call this a contribution.
>
> We have changed this contribution to reflect the new results we have added to the revised paper, however on this point we respectfully disagree as we feel it is still a valid contribution, albeit a small one. With regards to the joint sampling of sequence and structure as sampling from some $p(x,y)$, this is not a development of Anishchenko et al., it is one of the common ways of thinking about the protein design and structure prediction problems. Our sampling scheme is an autoregressive sample from DARK3 (always no refinement unless stated otherwise), effectively a forward pass, producing a sequence which is then fed to AlphaFold which is also a forward pass (ignoring any relaxation with OpenMM), producing an atomic structure.
>
> To quote Reviewer jJy9, they are effectively 'building blocks'; it's two models together that produce a sequence and a structure in seconds. Such a simple combination provides a flexible basis for further method development, and more practically, it rapidly provides atomic structures in `.pdb` files that can be immediately used with other ML and bioinformatics tools. By comparison, trDesign does not provide this flexibility. It performs a slow sequence optimization after which it has a sequence and predicted distance/orientation distributions. Following the typical trRosetta approach, it then needs to perform coarse-grained structure modeling with Rosetta's energy minimization protocols. This is then follow by all-atom structure refinement with Rosetta's FastRelax protocol. Our simple approach is fast neural network inference where as trDesigns approach includes significant Monte Carlo sampling and energy function minimization.
>
> >  ...also mention that annealing could in principle be replaced with gradient ascent by backpropagating to the sequence, akin to which others like [Killoran et al 2017] attempted in simpler settings, which seems more "neural" than what the submission is doing...
>
> Thank you, we agree this is interesting, and we initially explored this before submission and found it less promising. But we agree it is a valuable inclusion and we have added results for this gradient based approach, included as DARK$_0$-_Grad_ (Table 1).

---

> > ### Author Response · Authors · 2021-11-27
> > **Response to Reviewer sSh9 Cont.**
> >
> > Relevant to the following points, we have added further precise detail and background regarding _de novo_ design to the introduction. _De novo_ is unique in protein design and engineering as it's looking to use learned and first principles to design proteins. This is, loosely put, the underlying and general design question. This is very important for the relevance of our work; especially as to why we feel that demonstrating unconditional models that learn to generate novel sequences with ordered structures is important.
> >
> > > But, you don't need deep learning to sample from the space of protein sequences that are broken up into stretches of beta sheets and/or alpha helices.
> >
> > In the context of _de novo_ design this is not the case - this is our fault for not having clarified this. As above, we have added further background. Wih the exception of some specific classes of proteins (e.g. there is rich literature regarding rational design of coiled coils) designing all-alpha and all-beta structures can be incredibly difficult depending on the topology, especially for all-beta, and is yet to be done for many even common topologies. For example, A simple jellyroll topology (all-beta) was only recently achieved (Marcos et al., 2018), and a intramembrane beta-barrel was achieved this year (Vorobieva et al., 2021). There are certainly many challenging design cases with alpha-beta topologies but it is not accurate to characterize it as the most difficult (Woolfson, 2021). For example, the first de novo designed protein with a novel fold, Top7, was an alpha-beta protein (Kuhlman et al., 2003), and human players of the foldit videogame recently designed several (e.g. Foldit1, Peak6) (Koepnick et al., 2019).
> >
> > > In most protein design situations, there is obviously a design criterion ... go to the trouble of experimentally validating the secondary structures of predicted sequences, but they also do not express a clear vision in this regard.
> >
> > We have added to the text in various places further discussion regarding applications of our approach. Notably, we have also added an example of a novel design method developed with the aid of DARK that we call AlphaFold refinement.
> >
> > For a _de novo_ task, conditioning on a family of protein sequences and fine-tuning on them, or using their sequences or structures, in any direct way is 'cheating'. A huge amount of compute that goes into designing with methods like Rosetta, and producing sequence with structures predicted to be ordered is difficult. We speculate that Anishchenko et al didn't discuss this because being able to generate sequences with diverse and ordered structures in a _de novo_ setting, as we similarly do but much faster, is remarkable in and of itself. Almost all design problems require candidates with an ordered structure. Being able to guide that generation to particular ordered structures is, relatively, a much easier problem.
> >
> > With our approach it is easy to generate huge sets of sequences with predicted structures, and like we show with our example of an immunoglobin fold and our novel AlphaFold refinement method, this allows us to tackle high-interest design problems. Looking ahead though, there are straightforward next steps to explore making our approach more robust. For example, we can explore different controlled generation methods by bootstrapping AlphaFold predictions as pseudo-labels (e.g. conditioning on topologies).
> >
> > > By analogy to \beta-VAEs [Higgins et al 2017, ICLR], there might be a missed opportunity to tune the sample quality by introducing a simple hyperparameter that controls the weighting between the two KL terms.
> >
> > Thank you for this suggestion. We previously explored different weighting schemes between the two and did not find it to provide a clear benefit. We also found that the sequence prior provides a way to numerically check if a model collapses, although we did not see any collapse in our trained models. Thus, for clarity we do not explore it in the text, but it is still of interest for further work.
> >
> > > ...when comparing to other generative methods that had different goals and therefore different notions of 'quality'.
> >
> > Thank you, we have clarified this in the text. Would you be able to clarify your comment regarding what is meant by different goals? WRT the Good+ pLDDT results, we use pLDDT as a dual order-confidence metric because regardless of design method or task, candidates should be predicted confidently to have an ordered structure (unless the tasks is, unusually, to design a disordered protein).

---

> > > ### Author Response · Authors · 2021-11-27
> > > **Response to Reviewer sSh9 Cont.**
> > >
> > > > Could the authors please comment on whether other structures were observed in sampling... Do the samples depicted in Figure 2 represent, proportionally, the kind of samples generated?
> > >
> > > We did not include labels on Figure 2 (now Figure 5) but we have now included labels for each row, which we chose to have different classes of structure; the first row is all-alpha, the second is alpha and beta, the third is all beta. From performing the structure prediction of the DARK$_2$ training set we would expect DARK$_3$ to produce samples covering, at a minimum, more than a 1000 different topologies. There is a biasing towards all-a and all-b (together about 4/5th of samples) but the 1/5th of alpha-beta samples show significant diversity, unsurprisingly making up the vast majority of unique topologies seen. From a practical perspective, among all classes we see topologies like the immunoglobin fold (we also produce a design for one with our AlphaFold refinement method) that are of high scientific interest but yet to be designed.
> > >
> > > > I am surprised that the paper does not include a "DARK_0" baseline...
> > >
> > > Thank you, we agree this should have been included. We have added this to Table 2.
> > >
> > > > Since DARK includes multiple rounds, and each round includes both training and hill-climbing, it seems fair to ask whether applying the same computational budget to "annealing from purely random seeds" baseline would produce comparable results and diversity...
> > >
> > > We have added a significant amount of discussion regarding timing throughout the paper, as we feel this was very important and needed to be included. We include a discussion of this comparison in the first paragraph of 4.1. To summarize here, the full process of starting with nothing to having a trained DARK3 is around 10 times faster than sampling 500K samples. Sampling 500K sequences by annealing from purely random seeds would have been completely beyond available resources. From our experiments not included here, the same appears to be true from a 'hill-climbing from random seeds approach'. From the new results we have included showing the generality of our models we know that DARK iterations extend beyond the initial diversity of the 15K annealed samples. We can only speculate the diversity of 500K samples from an annealing method; DARK was in part built to make this quantity of sampling feasible.
> > >
> > > ---
> > >
> > > References:
> > >
> > > Kuhlman, B., Dantas, G., Ireton, G.C., Varani, G., Stoddard, B.L. and Baker, D., "Design of a novel globular protein fold with atomic-level accuracy." _Science_, (2003).
> > >
> > > Koepnick, B., Flatten, J., Husain, T., Ford, A., Silva, D.A., Bick, M.J., Bauer, A., Liu, G., Ishida, Y., Boykov, A. and Estep, R.D., "De novo protein design by citizen scientists." _Nature_, (2019).
> > >
> > > Marcos, E., Chidyausiku, T.M., McShan, A.C., Evangelidis, T., Nerli, S., Carter, L., Nivón, L.G., Davis, A., Oberdorfer, G., Tripsianes, K. and Sgourakis, N.G., "De novo design of a non-local β-sheet protein with high stability and accuracy." _Nature structural & molecular biology_, (2018).
> > >
> > > Vorobieva, A.A., White, P., Liang, B., Horne, J.E., Bera, A.K., Chow, C.M., Gerben, S., Marx, S., Kang, A., Stiving, A.Q. and Harvey, S.R. "De novo design of transmembrane β barrels" _Science_, (2021).
> > >
> > > Woolfson, D.N., "A brief history of de novo protein design: minimal, rational, and computational." _Journal of Molecular Biology_, (2021).

---

> > > > ### Comment · Reviewer_sSh9 · 2021-11-29
> > > > **On authors' comments**
> > > >
> > > > I have read the authors' comments and looked over the revisions. Many concerns were addressed.

---

### Official Review · Reviewer_Jhfe · 2021-12-03

**Correctness:** 2
**Technical Novelty And Significance:** 2
**Empirical Novelty And Significance:** 2
**Recommendation:** 3
**Confidence:** 4

**Main Review:**

Author's goal: Create a ML model that generates de novo sequences that fold.
- That fold could be anything at all. Simply "a folding predictor is confident that it folded my sequence accurately".

I think this is a fun idea, but too flawed. in application as it is a little circular  I am curious what the generated sequences look like... Not because the are likely to fold, rather because they are ones that "fool" AlphaFold.

Critique in summary:
The work boils down to an adversarial attack on DMSfold2, which proves to enrich for high scores in AlphaFold. But, adversarial attacks commonly create nonsense examples that exploit biases in the target model.
AlphaFold is not convincing validation, because it is too similar to the oracle used for optimization.
Both DMSfold2 and AlphaFold are misused. They've been trained & tested for multiple sequence alignments of natural sequences. There's very very little evidence that they accurately predict the fold of individual non-natural sequences.

The method and my problems with it:
1. Sequence generation = autoregressive sequence model. First, the AR model is trained on seed sequences. It then generates new sequences.
As a seed, the authors use sequences generated from trDesign (Baker lab 'hallucinations' model). Effectively, these are sequences predicted to fold according to Rosetta.

Problem = They've started with the answer, Rosetta designs.
- In the output, 922 / 950 seqs were ~33% identity to seed sequences.
- Need authors to include more detailed sequence comparison.
- That 33% might be enough 'cheating' to get high 'validation' scores, in which case they've just used Rosetta.
- They need to report the validation scores for the trDesign seed sequences too.
- Are the predicted folds for final output the same as for seed sequences?

2. Generated sequences are then mutated to optimize "folding score" = KL divergence of distograms (binned pair distances) according to DeepMetaPsiCov fold 2.
This should mean "optimize sequences so that predicted contact maps are highly non-random". With each iteration, the AR model is then trained on the optimized sequences plus the previous iterations' sequences.

Problem = They are optimizing a sequence generator to exploit a model OUTSIDE of its use-case. This is like an adversarial attack. I would expect to get sequences that exploit false biases.

3. For validation the final output sequences are run through AlphaFold prediction & scoring = pLDDT (predicted Local Distance Test score). Authors claim AlphaFold is independent validation from the loss function they optimized against.

Problem =
AlphaFold is not an independent validation, because it is virtually the same model, trained on the same data as DMSfold2, and outputs the same sort of score.
- Trains on MSAs
- Trains on natural sequences
- Very similar architectures
- Scores, AlphaFold pLDDT vs DMSfold2 confidence. Both are predictions of distogram measurements, LDDT and TM, respectively. And trained on PDB structures of natural sequences.
Also a misuse of AlphaFold the same way as in #1. AlphaFold uses MSAs, and is trained/validated 99% on natural sequences.


Results, boil down to 2 weak observations:
1. The basic training task is possible: training the AR model indeed creates sequences that improve KL Divergence of DMPfold2 distograms.
- Non-result.. should be guaranteed that the ML training will improve objective score.

2. The output sequences have higher AlphaFold predicted LDDT than random sequences, sequences output by an AR trained on UniRef50, or two other models.
- Comparative advantage is versus other unproven models, but not e.g. natural proteins, or Rosetta designs.
- Advantage is all but guaranteed by similarity to DMPfold2, which was optimized for.
- Unclear that the AlphaFold pLDDT score is meaningful for non-natural sequences.
- Unclear if higher scores are byproduct of using trDesign seed sequences.



**Summary Of The Paper:**

Dear Program Chair(s)
After reading the back and forth on this paper I feel it might help if i send comments _even though they are very very late


**Summary Of The Review:**

Author's goal: Create a ML model that generates de novo sequences that fold.
- That fold could be anything at all. Simply "a folding predictor is confident that it folded my sequence accurately".

I think this is a fun idea, but far too flawed. I am curious what the generated sequences look like... Not because the are likely to fold, rather because they are ones that "fool" AlphaFold.

Critique in summary:
The work boils down to an adversarial attack on DMSfold2, which proves to enrich for high scores in AlphaFold. But, adversarial attacks commonly create nonsense examples that exploit biases in the target model.
AlphaFold is not convincing validation, because it is too similar to the oracle used for optimization.
Both DMSfold2 and AlphaFold are misused. They've been trained & tested for multiple sequence alignments of natural sequences. There's very very little evidence that they accurately predict the fold of individual non-natural sequences.

The method and my problems with it:
1. Sequence generation = autoregressive sequence model. First, the AR model is trained on seed sequences. It then generates new sequences.
As a seed, the authors use sequences generated from trDesign (Baker lab 'hallucinations' model). Effectively, these are sequences predicted to fold according to Rosetta.

Problem = They've started with the answer, Rosetta designs.
- In the output, 922 / 950 seqs were ~33% identity to seed sequences.
- Need authors to include more detailed sequence comparison.
- That 33% might be enough 'cheating' to get high 'validation' scores, in which case they've just used Rosetta.
- They need to report the validation scores for the trDesign seed sequences too.
- Are the predicted folds for final output the same as for seed sequences?

2. Generated sequences are then mutated to optimize "folding score" = KL divergence of distograms (binned pair distances) according to DeepMetaPsiCov fold 2.
This should mean "optimize sequences so that predicted contact maps are highly non-random". With each iteration, the AR model is then trained on the optimized sequences plus the previous iterations' sequences.

Problem = They are optimizing a sequence generator to exploit a model OUTSIDE of its use-case. This is like an adversarial attack. I would expect to get sequences that exploit false biases.

3. For validation the final output sequences are run through AlphaFold prediction & scoring = pLDDT (predicted Local Distance Test score). Authors claim AlphaFold is independent validation from the loss function they optimized against.

Problem =
AlphaFold is not an independent validation, because it is virtually the same model, trained on the same data as DMSfold2, and outputs the same sort of score.
- Trains on MSAs
- Trains on natural sequences
- Very similar architectures
- Scores, AlphaFold pLDDT vs DMSfold2 confidence. Both are predictions of distogram measurements, LDDT and TM, respectively. And trained on PDB structures of natural sequences.
Also a misuse of AlphaFold the same way as in #1. AlphaFold uses MSAs, and is trained/validated 99% on natural sequences.


Results, boil down to 2 weak observations:
1. The basic training task is possible: training the AR model indeed creates sequences that improve KL Divergence of DMPfold2 distograms.
- Non-result.. should be guaranteed that the ML training will improve objective score.

2. The output sequences have higher AlphaFold predicted LDDT than random sequences, sequences output by an AR trained on UniRef50, or two other models.
- Comparative advantage is versus other unproven models, but not e.g. natural proteins, or Rosetta designs.
- Advantage is all but guaranteed by similarity to DMPfold2, which was optimized for.
- Unclear that the AlphaFold pLDDT score is meaningful for non-natural sequences.
- Unclear if higher scores are byproduct of using trDesign seed sequences.

---

> ### Comment · Area_Chair_duz9 · 2021-12-06
> **Review is far too late to be considered.**
>
> It is not fair to the authors for this review to be considered now, well beyond the deadline for first review, let alone for the discussion period itself.

---

### Author Response · Authors · 2021-11-27
**Overall Response to Reviewers**

We would like to thank all the reviewers for their time and effort evaluating our manuscript. We have tried to address all concerns raised. As a result we feel the paper has considerably improved, both from the addition of significant new methods, new result, and better communication of the work. Overall, we have made the following changes:

1. Significantly restructured the argument in the Introduction, to make our aims clearer, provide more background to _de novo_ design, and make our achievements hopefully clearer.
2. The abstract and contributions have been updated to reflect this.
3. Added further details to the Related Work regarding why natural sequence-based unconditional models fail to structurally generalize.
4. Clarified the involvement of trDesign and how we outperform it in the methods and Results.
5. Moved wcGAN and gramVAE results to the Appendix, and added DARK$_0$ pLDDT results, to clarify the results.
6. Moved sequence novelty results to the main text and added new results for performing the same analysis with the UF50 model.
7. Added methods and results directly showing that our models structurally generalize on a stringent unseen structure-based train-test set split.
8. Added results showing DARK iterations extend the training sets to include sequences with unseen structures.
9. Added significant new methods and results for a novel design method enabled by DARK, that we call AlphaFold Refinement, to provide further demonstrations of DARK models applications.
10. Included an example of using DARK and AlphaFold Refinement to design a high confidence candidate for an Immunoglobin fold, a fold yet to be successfully designed for and off high Biotech and medical interest.
11. Provided results showing how a trDesign-like approach struggles to extend to AlphaFold.
12. Added sampling speed and timing information throughout.
13. Added IG-score results for a gradient descent based sequence sampling method with DMPfold2, termed DARK$_0$-_Grad_.
14. Added Perplexity, IG-score, and Good+ pLDDT results (latter in the Appendix) for DARK$_1$-_Adversarial_, a model trained with DMPfold2 as an adversarial regularizer.
15. Added further details regarding the exclusion of cysteine.
16. Included further Appendix details to aid reproduction of the work.

We have responded in detail to all comments and we enthusiastically invite any further questions or discussions. We believe our work is of interest to researchers in both machine learning, computational biology, and the growing intersection between the two communities. Thank you again to the reviewers for your effort and time.

---

_Update 7th Dec_: As it is relevant and may be of interest to reviewers, the related work to ours, trDesign, has been recently [published in Nature with further results](https://www.nature.com/articles/s41586-021-04184-w) as _Anishchenko, I. et al. “De novo protein design by deep network hallucination.” Nature (2021)_.

---

### Decision · Program_Chairs · 2022-01-20

**Decision:**

Reject

**Comment:**

Despite a lively discussion and author explanation and revision, this paper remains below the bar for publication at ICLR. The technical exposition and goals remain poorly explained. The technical contribution is not sufficient. And the utility of the empirical results remain in question. The strong consensus among the reviewers who submitted reviews in a timely manner is that the paper is not suitable for publication.  The 5th and final review, was submitted too late, well beyond the end of the discussion period, and hence was not considered in this decision.